# SPAM: SPIKE-AWARE ADAM WITH MOMENTUM RESET FOR STABLE LLM TRAINING

Tianjin Huang[1,5], Ziquan Zhu[2], Gaojie Jin[1], Lu Liu[1], Zhangyang Wang[4], and Shiwei Liu[3,5]

[1]Department of Computer Science, University of Exeter, Exeter, UK
[2]Department of Computer Science, University of Leicester, Leicester, UK
[3]Mathematical Institute, University of Oxford, Oxford, UK
[4]Department of Electrical and Computer Engineering, The University of Texas at Austin, Austin, US
[5]Department of Mathematics and Computer Science, Eindhoven University of Technology, Eindhoven, NL

## ABSTRACT

Large Language Models (LLMs) have demonstrated exceptional performance across diverse tasks, yet their training remains highly resource-intensive and susceptible to critical challenges such as training instability. A predominant source of this instability stems from gradient and loss spikes, which disrupt the learning process, often leading to costly interventions like checkpoint recovery and experiment restarts, further amplifying inefficiencies. This paper presents a comprehensive investigation into gradient spikes observed during LLM training, revealing their prevalence across multiple architectures and datasets. Our analysis shows that these spikes can be up to $1000\times$ larger than typical gradients, substantially deteriorating model performance. To address this issue, we propose Spike-Aware Adam with Momentum Reset (SPAM), a novel optimizer designed to counteract gradient spikes through momentum reset and spike-aware gradient clipping. Extensive experiments, including both pre-training and fine-tuning, demonstrate that SPAM consistently surpasses Adam and its variants across a range of model scales. Additionally, SPAM facilitates memory-efficient training by enabling sparse momentum, where only a subset of momentum terms are maintained and updated. When operating under memory constraints, SPAM outperforms state-of-the-art memory-efficient optimizers such as GaLore and Adam-Mini. Our work underscores the importance of mitigating gradient spikes in LLM training and introduces an effective optimization strategy that enhances both training stability and resource efficiency at scale. Code is available at https://github.com/TianjinYellow/SPAM-Optimizer.git.

## 1 INTRODUCTION

Large Language Models (LLMs) have become fundamental in advancing state-of-the-art AI systems. Scaling LLMs, such as GPT-3 (Brown, 2020) and LLaMA (Touvron et al., 2023), has showcased unprecedented capabilities. However, training these large-scale models is fraught with challenges, particularly training instability. A major factor contributing to this instability is the occurrence of gradient and loss spikes during training, which disrupt the learning process at unpredictable intervals (Chowdhery et al., 2023; Zhang et al., 2022; Le Scao et al., 2023).

While architectural innovations have been proposed to mitigate these issues (Nguyen & Salazar, 2019; Shoeybi et al., 2019; Zeng et al., 2022; Ding et al., 2021; Wang et al., 2024; Dettmers et al., 2021; Scao et al., 2022; Takase et al., 2023), none can completely prevent the occurrence of spikes. In practice, the most widely adopted solution is to manually intervene by restarting training from a previous checkpoint and skipping data affected by the spike (Chowdhery et al., 2023). This method is resource-intensive, requiring frequent checkpoint saves, manual monitoring, and repeated experiment runs - all inefficient and undesirable.

Moreover, the sheer scale of LLMs necessitates vast computational resources. For example, training LLaMA required over 2048 A100-80GB GPUs (Touvron et al., 2023), posing significant environ-

mental and financial costs (Rillig et al., 2023; Patterson et al., 2021). These challenges highlight the need for more efficient training paradigms that reduce resource consumption without sacrificing performance.

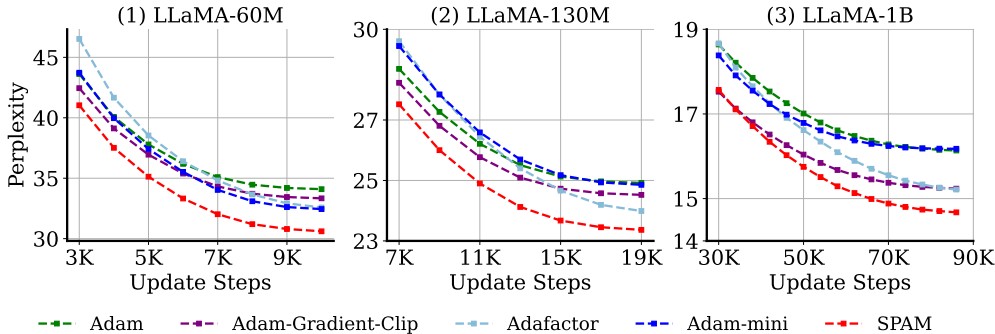

Figure 1: Perplexity of LLaMA models on C4 trained with various optimizers.

In this paper, we approach the issue from an optimization perspective rather than an architectural one. We first conduct an in-depth investigation of loss and gradient spikes during the training of various LLM architectures, spanning models from 60M to 1B parameters. Our study reveals several key observations:

- *Small yet frequent loss bumps:* Although catastrophic loss spikes are rare, we observe frequent small loss bumps that can easily be overlooked without close scrutiny.

- *Gradient spikes accompanying loss bumps:* These loss bumps, depiste small by their own, are consistently accompanied by significant gradient spikes, whose magnitudes can reach up to 1000 $\times$ greater than typical gradients. These spikes persist across layers, architectures, and datasets, even with established techniques applied.

- *Harmfulness of gradient spikes:* By nullifying the spiked gradients, we observe notable improvements in training performance, confirming that these spikes have a detrimental effect. Momentum-based optimizers, like Adam (Kingma, 2014; Loshchilov, 2017), suffer particularly from the accumulation of these spikes in their momentum terms, as we demonstrate both empirically and theoritically.

Inspired by these findings, we introduce **Sp**ike-Aware **A**dam with **M**omentum Reset (**SPAM**), an optimizer designed to counteract the negative effects of gradient spikes. SPAM introduces two key innovations: (1) periodic reset of the first and second moments to eliminate the harmful accumulation of spiked gradients, and (2) identification and adaptive re-scaling of spiked gradients to manageable levels, preserving their directional information while mitigating their magnitude. We validate SPAM through extensive experiments, demonstrating its superior performance across various LLM sizes in both pre-training and fine-tuning tasks.

Furthermore, momentum reset enables the development of sparse momentum, where only a selected subset of momentum terms is computed and stored during training, drastically reducing memory costs. Our results show that SPAM surpasses leading memory-efficient optimizers such as GaLore (Zhao et al., 2024) and Adam-Mini (Zhang et al., 2024a) with good margins, even under memory constraints.

**Summary of Contributions:**

⋆ Comprehensive analysis of gradient spikes across multiple LLM architectures, revealing their significant impact on training stability and performance.

⋆ Introduction of SPAM, a novel optimizer with momentum reset and spike-aware clipping that outperforms existing methods like Adam and Adafactor.

⋆ A memory-efficient version of SPAM that leverages sparse momentum to reduce memory usage while maintaining superior performance compared to state-of-the-art memory-efficient optimizers.

## 2 GRADIENT SPIKES

In this section, we formally define gradient spikes and then present the intriguing findings from our investigation into the training loss and gradient dynamics during LLM training.

Gradient spikes refer to a phenomenon that occurs during training where the magnitude of certain gradients significantly exceeds their historical values. To more precisely identify and analyze instances of gradient spikes, we introduce the Gradient Spike Score as a measurement of the deviation of a gradient's magnitude from its typical behavior over time. By quantifying this relative change, we can monitor the dynamics of gradients during training.

**Definition 2.1** (**Gradient Spike Score**). *Let $\{g_0, g_1, \ldots, g_{T-1}, g_T\}$ be the sequence of gradient obtained during the training process from time step $0$ to $T$. The Spike Score of the gradient at the $i^{th}$ step, denoted as $\mathrm{GSS}(g_i)$, is defined as the ratio of the magnitude of the gradient at that step to the average magnitude of the gradients across all steps:*

$$\mathrm{GSS}(g_i) = \frac{|g_i|}{\frac{1}{T+1}\sum_{j=0}^{T}|g_j|}$$

A gradient $g_i$ is considered a spiked gradient if its $\mathrm{GSS}(g_i)$ exceeds a predetermined threshold $\theta$, i.e., $\mathrm{GSS}(g_i) > \theta$ indicating a significant increase from typical fluctuations, often amounting to increases of two or three orders of magnitude.

### 2.1 PRESENCE OF GRADIENT SPIKES DURING LLM TRAINING

Building upon the above concepts, we further explore the presence of gradient spikes during LLM training. Specifically, we monitor the gradients of the entire model over the initial $1,000$ training steps and identify gradient spikes using the condition $\mathrm{GSS}(g_i) > 50$. Our investigation encompasses two widely adopted LLM architectures, LLaMA (Touvron et al., 2023)[1] and Pythia (Biderman et al., 2023), with model sizes varying from 60M to 1B parameters. Experiments were conducted on two datasets: the well-known C4 dataset (Raffel et al., 2020) and a cleaner high-quality dataset, SlimPajama (Soboleva et al., 2023). Please refer to Appendix D for more details. Our key observations can be summarized as follows:

① **Loss bumps accompanying gradient spikes occur irregularly during LLM training.** Although we do not observe severe loss spikes that lead to catastrophic divergence (Takase et al., 2023; Chowdhery et al., 2023), we do observe subtle loss bumps that happen quite frequently. For instance, Figure 2-top illustrates the training loss of LLaMA-60M, 350M, and 1B models, where several loss bumps can be seen during training, marked with red circles. We further investigate the model's gradients at these moments and observe that gradient spikes coincide with the loss bumps, as demonstrated in Figure 2-bottom. While gradients remain small for most of the training, they suddenly become extremely large when loss spikes occur.

② **Gradient spikes are widely presented in different layers, across different architectures, model sizes, and datasets.** Overall, we observed many gradient spikes across all layer types, as detailed in Figure 3-(4) and Appendix A & B, with LayerNorm layers, in particular, experiencing an exceptionally high frequency of spikes. Figure 2 demonstrates that models of varying sizes, from 60M to 1B, all exhibit gradient spikes. To verify whether architecture is the root cause of these spikes, we conducted experiments with Pythia-70M, which also suffers from numerous gradient anomalies, as shown in Figure 3. Additionally, we found that gradient spikes occur even when using cleaner, high-quality datasets such as SlimPajama, although the frequency of spikes is reduced with this cleaner dataset.

③ **Advanced spike mitigation approaches cannot completely eliminate gradient spikes.** We also evaluate whether previously proposed techniques for addressing spikes can eliminate gradient spikes. Specifically, we assess multiple approaches, including Scaled Initialization (Nguyen & Salazar, 2019; Shoeybi et al., 2019), Embed LN (Dettmers et al., 2021), Scaled Embed (Takase et al., 2023), and Embed Detach (Zeng et al., 2022). The results in Figure 4 show that while some approaches perform better than others, they cannot completely eliminate gradient spikes. More specifically, we find that Scaled Embed and Embed LN significantly reduce the number of gradient

---

[1]We adopt the LLaMa models used in Lialin et al. (2023b); Zhao et al. (2024).

spikes, while the other methods offer little to no improvement, consistent with the findings reported in Takase et al. (2023).

Our observation of loss bumps likely relates to the edge of stability (EoS) phenomenon (Cohen et al., 2021), where the sharpness of the network hovers near the stability threshold for the remainder of training while the loss continues to decrease, albeit non-monotonically. However, the EoS phenomenon has not been extensively studied at the scale of LLMs. Moreover, our study reveals that these loss bumps have harmful effects on LLM training, which were not observed in previous studies.

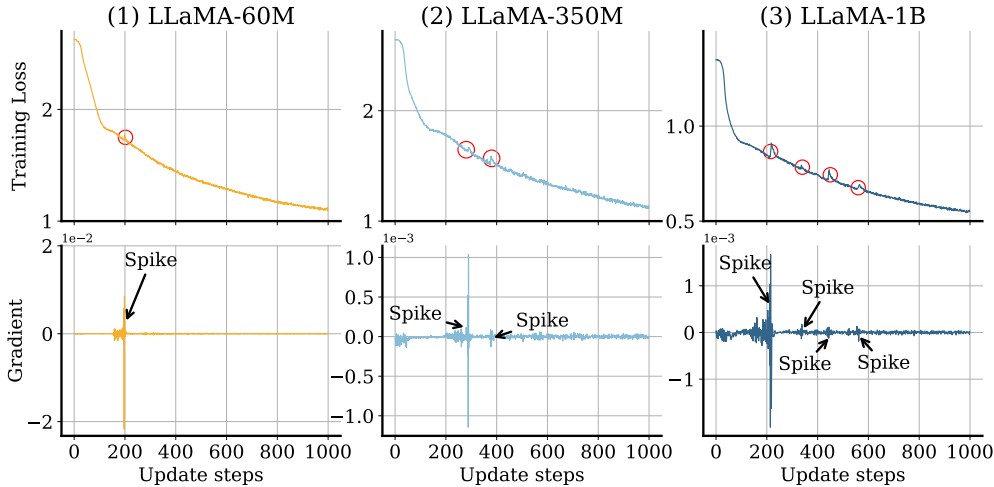

Figure 2: **Training loss lumps and their corresponding gradient spikes.** Gradient trajectories are collected with LLaMa-60M, 350M, 1B models on C4 datasets. Gradient spikes are detected using $\text{GSS}(g_i) > 50$.

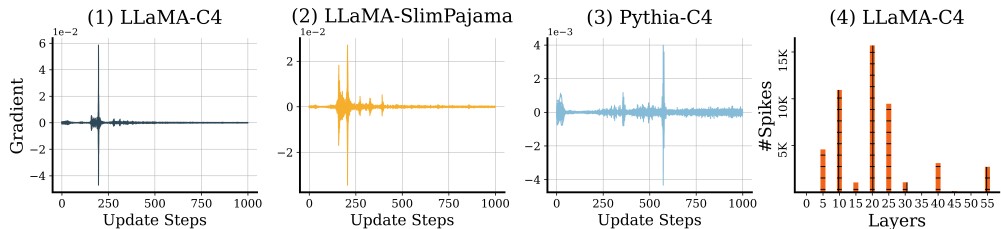

Figure 3: **Spike gradients present across different architectures and datasets.** $(1) - (3)$: Plots of 100 randomly selected spike gradients (using $\text{GSS}(g_i) > 50$) of LLaMa-60M and Pythia-70M on C4 and SlimPajama datasets. (4): Number of spiked gradients every 5 layers during the first 1K steps in LLaMa-60M on C4.

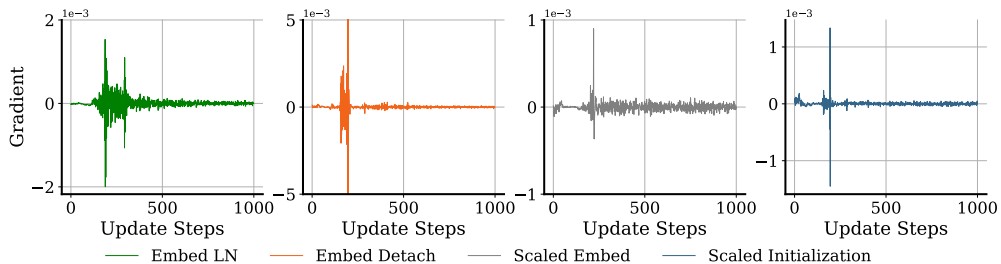

Figure 4: **Advanced spike mitigation approaches can not completely eliminate gradient spikes.** Gradient trajectories are collected with LLaMa-60M on C4. The spike gradient is detected via $\text{GSS}(g_i) > 50$.

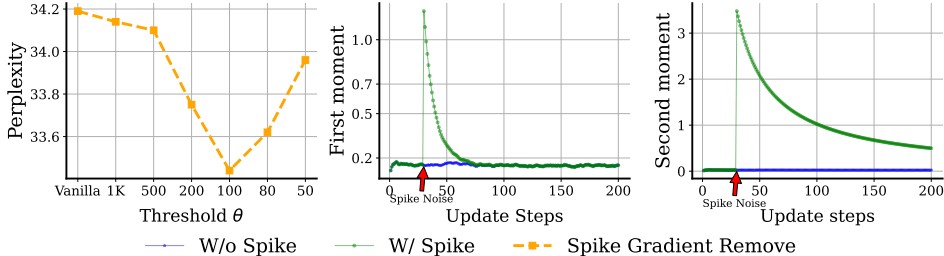

Figure 5: **Left**: Perplexity of the final model after zeroing out spiked gradients using various $\theta$, $\mathrm{GSS}(g_i) > \theta$. Experiments are conducted using LLaMa-60M on C4. **Middle and Right**: Impact of spiked Gradients on the first and second Moments. Simulated gradients ($g_i \sim \mathcal{N}(\mu, \sigma^2)$) are used to visualize the prolonged effects of gradient spikes on the first and second moments, with a large spike noise introduced at the 30th step.

## 2.2 EFFECTS OF GRADIENT SPIKES ON LLM TRAINING

After identifying the presence of gradient spikes during training, a crucial question arises: are these gradient spikes detrimental or, perhaps counterintuitively, beneficial to the training of LLMs? To address this, we conducted a series of experiments as follows. Our findings confirm that gradient spikes are indeed harmful to LLM training, exerting prolonged negative effects on both the first and second moments, as discussed below.

**Gradient spikes negatively impact LLM training.** One direct way to assess the impact of gradient spikes is by nullifying the spiked gradients during training and observing the final training performance. We first detect spiked gradients using various thresholds $\theta$ and then set those gradients to zero. Figure 5-Left reports the results of LLaMA-60M on C4. Surprisingly, zeroing out these spiked gradients leads to improved model performance, evidenced by a reduction in perplexity. This observation clearly indicates that gradient spikes hinder effective training, and their removal is beneficial to overall model performance.

**Gradient spikes have prolonged detrimental effects on the first and second moments.** Due to the exponential averaging of the momentum mechanism, the influence of a gradient spike decays slowly over time. To demonstrate this, we conduct a simulation experiment using Adam. In this experiment, we model the gradients as random variables drawn from a Gaussian distribution with mean $\mu = 0.1$ and variance $\sigma^2 = 0.1$, i.e., $g_i \sim \mathcal{N}(\mu, \sigma^2)$. We sample gradients and track their corresponding moments over 200 steps, introducing a gradient spike at step 30 with a large magnitude of 10. As shown in Figure 5-Middle and Right, the spike's amplification persists, influencing both moments across subsequent steps. For example, it takes approximately 50 steps for the first moment to recover from the spike, while the second moment takes significantly longer, with the effect persisting beyond 200 steps. Two key factors plausibly contribute to this difference: (1) the second moment typically employs a larger exponential decay rate than the first (0.999 vs. 0.9); and (2) the second moment depends on the squared gradients, making it more sensitive to large spikes.

## 2.3 PRELINMINARY ANALYSIS WITH THEORY IMPLICATIONS

We hereby provide a very preliminary analysis to help probe *why gradient spikes have a significant impact on the regret bound of Adam-like algorithms*. We strictly follow the setting and notations used in Alacaoglu et al. (2020). Specifically, referring to **Theorem 1** in the paper, the regret bound consists of two main terms:

$$R(T) \le \frac{D^2\sqrt{T}}{2\alpha(1-\beta_1)} \sum_{i=1}^{d} \hat{v}_{T,i}^{1/2} + \frac{\alpha\sqrt{1+\log T}}{\sqrt{(1-\beta_2)(1-\gamma)}} \sum_{i=1}^{d} \sqrt{\sum_{t=1}^{T} g_{t,i}^2},$$

where $\gamma = \frac{\beta_1^2}{\beta_2}$. Gradient spikes directly affect these terms by increasing the magnitudes of the gradients $g_t$. In their **Lemma 3**, it is shown that the norm $\|m_t\|_{\hat{v}_t^{-1/2}}^2$ depends on the accumulated

gradients:

$$\|m_t\|_{\hat{v}_t^{-1/2}}^2 \leq \frac{(1-\beta_1)^2}{\sqrt{(1-\beta_2)(1-\gamma)}} \sum_{i=1}^{d} \sum_{j=1}^{t} \beta_1^{t-j}|g_{j,i}|.$$

When gradient spikes occur, the values of $g_{j,i}$ become significantly larger for some $j$ and $i$, which in turn increases the bound on $\|m_t\|_{\hat{v}_t^{-1/2}}^2$. This enlargement propagates through the analysis, particularly affecting the accumulation term $\sum_{t=1}^{T} \alpha_t \|m_t\|_{\hat{v}_t^{-1/2}}^2$ in their **Lemma 4**, which is bounded by:

$$\sum_{t=1}^{T} \alpha_t \|m_t\|_{\hat{v}_t^{-1/2}}^2 \leq \frac{(1-\beta_1)\alpha\sqrt{1+\log T}}{\sqrt{(1-\beta_2)(1-\gamma)}} \sum_{i=1}^{d} \sqrt{\sum_{t=1}^{T} g_{t,i}^2}.$$

Here, gradient spikes increase $\sum_{t=1}^{T} g_{t,i}^2$ significantly, especially in the coordinates where the spikes occur, leading to a larger bound.

Finally, in the main regret bound (**Equation (9)** in the paper), these enlarged terms result in a looser (larger) overall regret bound due to the presence of gradient spikes. The increased $\hat{v}_{T,i}^{1/2}$ and $\sum_{t=1}^{T} g_{t,i}^2$ directly contribute to the regret bound becoming less tight. This theoretical implication highlights that while adaptive algorithms like AMSGRAD adjust learning rates based on gradient history, they may perform worse in terms of regret when large gradient spikes are present due to the increased cumulative squared gradients and decreased effective learning rate.

It is important to note that our goal is **not** to claim theoretical innovations, but rather to quantitatively assess how gradient spikes degrade Adam-like optimization, and that is only explored in a very limited context. We would like to clarify the limitations of this analysis: (1) The analysis assumes convexity, which may not apply in non-convex settings (but is often mitigated by assuming Polyak-Lojasiewicz condition or so). (2) The assumption $\|g_t\|_\infty \leq G$, where $G$ denotes the maximum allowable gradient bound, may be in conflict with the presence of gradient spikes if $G$ is not sufficiently large to capture them. (3) There is a significant dependence on $G$, and if $G$ is set too high to accommodate spikes, the constants in the regret bound grow disproportionately, potentially making the bound meaningless. Nonetheless, we find that our analysis aligns well with our experimental results, and we leave a more rigorous theoretical exploration for future work.

## 3 SPIKE-AWARE ADAM WITH WITH MOMENTUM RESET (SPAM)

In this section, we introduce Spike-Aware Adam with Momentum Reset (SPAM). Unlike previous solutions that introduce architectural innovations to mitigate the decremental effects of gradient spikes (Nguyen & Salazar, 2019; Zeng et al., 2022; Dettmers et al., 2021; Takase et al., 2023), we attempt to address this issue from an optimization perspective. Concretely, we integrate Momentum Reset and Spike-Aware Clipping into Adam to deal with gradient spikes. In addition, we introduce a memory-efficient version of SPAM, which incorporates Sparse Momentum, significantly reducing the memory footprint during LLM training. Pseudocode of SPAM is in Algorithm 1.

**Momentum Reset.** To mitigate the detrimental effects of gradient spikes on training stability, we introduce Momentum Reset. Momentum Reset involves periodically resetting the accumulated first and second moments used by adaptive optimizers such as Adam. These optimizers rely on exponential moving averages of past gradients to inform parameter updates. However, when a gradient spike occurs, it can significantly inflate these moments, causing the impact of the spike to persist over many subsequent iterations. By resetting the momentum terms at regular intervals of $\Delta T$ training iterations, we can prevent the lingering influence of anomalously large gradients on the optimizer's state. This practice ensures that parameter updates are based on recent, more normal gradients rather than being skewed by gradient spikes. To mitigate potential instability caused by momentum reset, we perform $N$ steps ($N = 150$ by default) of cosine warmup following each reset operation.

**Spike-Aware Clipping.** To further mitigate gradient spikes during intervals, we introduce Spike-Aware Clipping. While our initial experiments indicate that setting spiked gradients to zero can enhance performance, this approach completely removes the learning signal for those parameters, including valuable directional information critical to the optimization process. To address this, SPAM

identifies gradients that exceed a predefined threshold $\theta$ and scales them to a manageable value, preserving their directional information while controlling their magnitude.

Detecting gradient spikes using GSS defined in Definition 2.1 would require knowing and storing all gradients in advance—a method that is impractical for LLM training due to memory constraints. We adopt a more memory-efficient, on-the-fly approach by leveraging the components already calculated by Adam. Formally, we detect gradient spikes by identifying gradients $g_i$ that meet the following condition: $\mathcal{G} = \left\{ g_i \mid \frac{g_i^2}{V_i} > \theta \right\}$ where $V_i$ is the second moment of Adam and $\theta$ is the threshold used for the approximate GSS $= \frac{g_i^2}{V_i}$. Note that we only use GSS defined in Definition 2.1 for the gradient spike analysis in Section 2. For real training, we employ the above approximation version. Since $V_i$ is essentially the moving average of $g_i^2$, this method efficiently identifies spikes without incurring additional overhead or the need to store the entire gradient history. Once detected, these spikes are clipped by scaling them to a manageable value. Specifically, for each spike gradient, we apply the operation: $g_i = \text{sign}(g_i) \cdot \sqrt{\theta V_i}$. This technique is particularly useful when combined with Momentum Reset. By incorporating these strategies, SPAM effectively mitigates the negative impact of gradient spikes, improving training stability and performance.

Note that unlike the Update Clipping used in Adafactor (Shazeer & Stern, 2018), which is applied to the whole weight update matrix when its Root Mean Square is larger than 1, our spike-aware clipping is directly applied to the spiked gradients $g_i$ whose magnitudes are significantly larger than its $\sqrt{v_i}$, e.g., $> 50\times$.

**Sparse Momentum.** Momentum reset paves the way for the development of sparse momentum, a technique designed to reduce memory usage and computation during the training of LLMs. In traditional momentum-based optimizers, such as Adam, momentum is updated and stored for all parameters, which can be memory-intensive for large-scale models. Sparse momentum offers a more memory-efficient alternative by updating and maintaining only a dynamically selected subset of moments at each iteration. The percentange of selected subset is denoted by $\%d$.

Key questions surrounding sparse momentum include how to effectively select parameter subsets, how to determine the sampling frequency, and whether to retain momentum for weights that are sampled consecutively . Our empirical analysis shows that random sampling is the most effective strategy for selecting subsets of parameters. For the other questions, we find that they align well with the momentum reset strategy. Specifically, setting the sampling frequency to match the momentum reset frequency, and resetting the momentum of all weights, even when they are sampled consecutively, yield the most robust results.

Table 1: Comparison with various optimizers on pre-training various sizes of LLaMA models on C4. Perplexity is reported.

| Model Size | 60M | 130M | 350M | 1B |
|---|---|---|---|---|
| Adam-mini | 34.10 | 24.85 | 19.05 | 16.07 |
| Adam | 34.09 | 24.91 | 18.77 | 16.13 |
| Adam+Gradient-Clip-Value | 33.65 | 24.72 | 18.52 | 15.77 |
| Adam+Gradient-Clip-Norm | 33.33 | 24.88 | 18.51 | 15.22 |
| Adafactor | 32.57 | 23.98 | 17.74 | 15.19 |
| SPAM | **30.46** | **23.36** | **17.42** | **14.66** |
| Training Tokens | 1.1B | 2.2B | 6.4B | 11.6B |

Table 2: Perplexity of Applying Advanced Techniques on LLaMA-60M.

| Optimizer | Perplexity |
|---|---|
| Adam | 34.09 |
| Adam+Embed LN | 33.61 |
| Adam+Embed Detach | 34.48 |
| Adam+Scaled Embed | 33.87 |
| Adam+Scaled Initalization | 34.29 |
| SPAM | **30.46** |

## 4 EXPERIMENTS

To demonstrate the efficacy of our proposed method, we conduct experiments on both pre-training and supervised fine-tuning using various sizes of the LLaMA model on C4 dataset.

**Baselines.** We adopt several widely-used optimizers as our baselines. Since SPAM is built upon Adam, Adam serves as our most direct baseline. We also incorporate two common gradient clipping approaches with Adam: (1) Value Clip, which clips all gradients when their absolute value exceeds a threshold; and (2) Norm Clip, which scales the entire gradient if the L2 norm of the gradient vector exceeds a certain threshold. Additionally, we compare against another widely-used optimizer, Adafactor (Shazeer & Stern, 2018). In terms of spike mitigation techniques, we eval-

uate `SPAM` against previous approaches, including Scaled Initialization (Nguyen & Salazar, 2019; Shoeybi et al., 2019), Embed LN (Dettmers et al., 2021), Scaled Embed (Takase et al., 2023), and Embed Detach (Zeng et al., 2022). For memory-efficient optimization methods, we include Adam-Mini (Zhang et al., 2024a), Galore (Zhao et al., 2024), LoRA (Hu et al., 2021), and ReLoRA (Lialin et al., 2023a).

**Architecture and hyperparameters.** Following (Lialin et al., 2023a; Zhao et al., 2024), we conduct our experiments using the LLaMA-based architecture with various sizes from 60M to 1B parameters, incorporating RMSNorm (Shazeer, 2020) and SwiGLU activations (Zhang & Sennrich, 2019). For each model size, we use the same set of hyperparameters across methods, varying only the learning rate, where we sweep over a set of learning rates from $1e-4$ to $1e-3$, incrementing by $2e-4$ for each optimizer. All experiments are conducted using the BF16 format. We set clip threshold as $1$ and $1e-3$ for Norm Clip and Value Clip, respectively, following the setting in Takase et al. (2023). We set hyper-parameters for Adafactor following the original paper (Shazeer & Stern, 2018) where $\epsilon_1 = 10^{-30}, \epsilon_2 = 10^{-3}$ and $d = 1.0$. For `SPAM`, we set reset intervals $\Delta T = 500$, lr warmup step $N = 150$ and GSS threshold $\theta = 5000$. Detailed descriptions of our task setups and hyperparameters are provided in the Appendix D.

## 4.1 PERFORMANCE OF LLM PRE-TRAINING

**Pre-training.** We report the training curves of various LLaMA models on the C4 dataset as well as the final perplexity in Figure 1 and Table 1, respectively. Overall, we observe that `SPAM` consistently achieves superior performance. As a memory-efficient approach, Adam-mini performs on par with Adam, consistent with the results reported in Zhang et al. (2024a). Commonly used gradient clipping techniques such as Value Clip and Norm Clip improve performance over Adam, with the latter achieving slightly better results. Adafactor further outperforms the aforementioned approaches, demonstrating its effectiveness. `SPAM` consistently outperforms all baselines across various LLaMA model sizes, highlighting the benefits of integrating momentum reset and spike-aware clipping techniques. All spike mitigation approaches fall short of `SPAM` as shown in Table 2. Additionally, Appendix E shows that `SPAM` can perform on par with or better than Adam in vision tasks and Appendix G shows that `SPAM` outperforms Adam in time series forecasting tasks.

Table 3: Comparison with memory-efficient algorithms on pre-training various sizes of LLaMA models on C4 dataset. Validation perplexity is reported, along with a memory estimate of the total of parameters, optimizer states based on BF16 format.The results of GaLore, Full-Rank, LoRA and ReLoRA are obtained from Zhao et al. (2024).

|  | 60**M** | 130**M** | 350**M** | 1**B** |
|---|---|---|---|---|
| Adam | 34.06 (0.36G) | 25.08 (0.76G) | 18.80 (2.06G) | 15.56 (7.80G) |
| ReLoRA | 37.04 (0.36G) | 29.37 (0.80G) | 29.08 (1.76G) | 18.33 (6.17G) |
| LoRA | 34.99 (0.26G) | 33.92 (0.54G) | 25.58 (1.08G) | 19.21 (6.17G) |
| GaLore | 34.88 (0.24G) | 25.36 (0.52G) | 18.95 (1.22G) | 15.64 (4.38G) |
| SPAM | **32.39** (0.24G) | **23.98** (0.52G) | **18.28** (1.22G) | **15.60** (4.38G) |
| Training Tokens | 1.1B | 2.2B | 6.4B | 11.6B |

**Memory-efficient Pre-training.** We evaluate `SPAM` by specifying $d\%$ such that its memory usage, including both parameters and optimizer states, matches that of Galore. For Galore, LoRA, and ReLoRA baselines, we set the ranks $r = 128, 256, 256, 512$ for the 60M, 130M, 350M, and 1B models, respectively, following the setup in Galore (Zhao et al., 2024). The results in Table 3 show that `SPAM` consistently outperforms all the baselines by a good margin, demonstrating its effectiveness as a memory-efficient optimizer.

## 4.2 PERFORMANCE OF LLM FINE-TUNING

In this section, we evaluate the effectiveness of `SPAM` for supervised fine-tuning. Following Li et al. (2024a), we fine-tune LLaMA2-7B on Commonsense170K (Hu et al., 2023) and test on 8 downstream tasks. We do not apply layer-wise weight updates for GaLore and `SPAM`. The rank is set to 8 for all low-rank baselines. Correspondingly, the density of `SPAM` is set to $0.25\%$ to maintain a comparable memory cost. The results are reported in Table 4. We observe that `SPAM` substantially outperforms other memory-efficient methods, exceeding full fine-tuning by a notable margin.

Table 4: Fine-tuning performance of LLaMa2-7B on various downstream tasks. The "Mem." denotes the running GPU memory. The mean and standard deviation of 10 repeated experiments are reported.

| Method | Mem. | BoolQ | PIQA | SIQA | HellaSwag | WinoGrande | ARC-e | ARC-c | OBQA | Avg. |
|---|---|---|---|---|---|---|---|---|---|---|
| Adam (Full FT) | 61G | 79.7±0.1 | 79.1±0.1 | 51.3±0.05 | 58.5±0.02 | 74.8±0.2 | 79.2±0.1 | 48.2±0.01 | 36.2±0.2 | 63.4±0.1 |
| LoRA | 26G | 75.8±0.4 | 79.0±0.1 | 56.3±0.1 | 59.9±0.04 | 79.6±0.2 | 77.6±0.1 | 46.9±0.1 | 34.4±0.3 | 63.7±0.2 |
| GaLore | 36G | 82.8±0.7 | 78.4±0.2 | 55.8±0.4 | 56.3±0.5 | 79.0±0.1 | 75.9±0.4 | 46.2±0.5 | 34.2±0.1 | 63.6±0.4 |
| SPAM ($d = 0.25\%$) | 36G | 85.0±0.2 | 78.9±0.2 | 55.7±0.2 | 57.8±0.1 | 78.9±0.2 | 76.5±0.2 | 47.3±0.2 | 35.1±0.3 | 64.4±0.2 |
| SPAM ($d = 100\%$) | 61G | 87.1±0.2 | 79.5±0.1 | 58.3±0.1 | 58.1±0.04 | 83.3±0.2 | 79.2±0.2 | 48.6±0.1 | 40.1±0.2 | 66.7±0.1 |

# 5 ABLATION STUDY

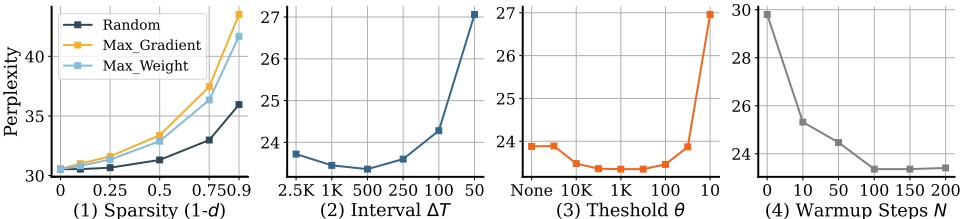

Figure 6: Ablations for sparse subset selection strategy, momentum reset inteval, GSS threshold and warmup steps. "None" denote that the spike-aware clipping is not applied.

**Selection strategy for sparse momentum.** Many strategies have been proposed to select subsets of parameters for sparse training, such as random selection (Liu et al., 2022a), max weight magnitude (Mocanu et al., 2018), and max gradient magnitude (Evci et al., 2020). Among these strategies, the most effective approach for sparse momentum training remains unclear. To investigate this, we conduct experiments with LLaMA-60M on the C4 dataset. The results are reported in Figure 6-(1). Interestingly, we find that randomly selecting subsets of parameters performs significantly better than the other two strategies for our sparse momentum. One plausible explanation for this discrepancy is that random selection allows for rapid exploration across all model parameters, whereas gradient- or weight-based strategies might be confined to the same subset of parameters during training.

**Momentum reset interval $\Delta T$.** To investigate the impact of interval $\Delta T$, we conduct experiments based on LLaMA-130M and C4 with varying $\Delta T$ fromm 50 to 2500. The warmup steps is set to 150 and the threshhold $\theta$ is set to 5000. The results are reported in Figure 6-(2). We observe a performance improvement as the interval $\Delta T$ decreases from 2500 to 500. However, when $\Delta T$ is further shortened, performance begins to degrade. This suggests that while momentum resets can enhance performance, excessively frequent resets may be detrimental to overall results.

**GSS threshold $\theta$.** Threshold $\theta$ decides which gradient are detected as spikes. To illustrate the impact of $\theta$ on SPAM, we present the results of LLaMA-130M in Figure 6-(3) with varying $\theta$ from 20000 to 10. The warmup steps is set to 150 and the interval $\Delta T$ is set to 500. We observe that performance improves as $\theta$ is reduced from extremely large values to smaller values, such as 1000, indicating that spike gradient clipping and momentum reset techniques have a mutually reinforcing effect. However, excessively small $\theta$ may interfere with the true gradient, ultimately leading to a degradation in performance.

**Warmup steps $N$.** We assess the impact of the warmup procedure following each momentum reset by presenting the performance of LLaMA-130M with different warmup steps, ranging from 0 to 200, in Figure 6-(4). The results indicate a significant performance drop when no warmup is applied ($N = 0$), compared to when a warmup is used. In addition, performance reach to optimal when the warmup duration is set to approximately 150 steps.

# 6 RELATED WORK

**Instability of Training Large Language Models.** LLMs are well-known for their training instability (Molybog et al., 2023), often experiencing irregular loss spikes that can lead to catastrophic divergence (Chowdhery et al., 2023). To address this issue, researchers have developed various sta-

bilization techniques. While we outline several key approaches, we acknowledge that this overview may not cover all significant contributions in the field.

One prominent approach involves architectural modifications. Xiong et al. (2020) demonstrated that using Post-LN in Transformers leads to larger gradients near the output layer, resulting in training instability, especially with large learning rates. In contrast, Pre-LN helps maintain well-behaved gradients during initialization, promoting more stable training. *Embed LN*, introduced by Dettmers et al. (2021), adds an additional LayerNorm after the embedding layer to improve stability, though it may cause performance degradation, as noted by Scao et al. (2022). *Embed Detach*, proposed by Ding et al. (2021) and further extended by Zeng et al. (2022) for LLMs, addresses loss spikes by shrinking embedding gradients. *DeepNorm*, developed by Wang et al. (2024), enhances stability in deep Transformers by scaling up the residual connection before applying LayerNorm. Additionally, $\alpha$*Reparam* (Zhai et al., 2023) re-parameterizes all linear layers using spectral normalization to prevent attention entropy collapse.

Another set of approaches focuses on improving initialization to mitigate training instability. *Scaled Embed*, proposed by Takase et al. (2023), scales up embeddings to stabilize LayerNorm gradients. *Scaled Initialization* (Nguyen & Salazar, 2019) introduces a parameter initialization strategy using a smaller normal distribution $\mathcal{N}(0, \sqrt{2/5d}/\sqrt{2N})$ to stabilize training dynamics. Additionally, *Fixup* (Zhang et al., 2019; Huang et al., 2020) claims that proper initialization can entirely eliminate the need for LayerNorm.

Very recently, Sun et al. (2025) introduced the concept of "the Curse of Depth", emphasizing that nearly half of the deep layers in modern LLMs underperform relative to expectations. They attribute this issue to the widespread adoption of Pre-LN (Li et al., 2024b) and propose scaling down the LayerNorm output as an effective solution.

**Momentum Reset.** Momentum reset is not a new approach. It has been used in Gu et al. (2013); Nesterov (2013) to solve the rippling behavior of Nesterov's Accelerated Gradient (NAG) (Nesterov, 1983) in the high-momentum regime, particularly in the context of convex optimization problems. O'donoghue & Candes (2015) further proposed adaptive reset where the momentum will be reset when an increase in the function value is observed. Unlike these earlier work, we leverage momentum reset to mitigate the detrimental effects of gradient spikes that arise during the training of billion-parameter language models, which present a large-scale, non-convex optimization challenge.

**Memory-Efficient Optimizers.** There have been several efforts to reduce Adam's memory footprint. SM3 (Anil et al., 2019), a lightweight variant of AdaGrad (Duchi et al., 2011), selects the learning rate for the $i$-th parameter by taking the minimum value from a set of candidates, each associated with the maximum squared gradient under a predetermined cover. Adafactor (Shazeer & Stern, 2018) and its variant CAME (Luo et al., 2023) utilize non-negative low-rank factorization over Adam's second-moment estimate, $v$. Adam-mini (Zhang et al., 2024a) partitions the parameters into blocks and assigns a single learning rate $v$ to each block to reduce memory. Similar approaches were proposed in (Zheng & Kwok, 2019; Ginsburg et al., 2019). Low-precision optimizers are studied in (Dettmers et al., 2021). Recently, GaLore (Zhao et al., 2024; Zhang et al., 2024b) enables the full-parameter training of LLMs through low-rank gradient updates.

## 7 CONCLUSION

In this paper, we presented a comprehensive study of gradient and loss spikes in LLM training, demonstrating their detrimental impact on training stability and performance across a variety of architectures and datasets. To address this issue, we propose Spike-Aware Adam with Momentum Reset (`SPAM`), a novel optimizer designed to counteract gradient spikes through momentum reset and spike-aware gradient clipping. The effectiveness of `SPAM` is backed up with extensive experiments across various LLM model sizes, where `SPAM` consistently outperformed Adam and other state-of-the-art optimizers by a good margin. When operating under memory constraints, `SPAM` motivates the feasibility of sparse momentum training, outperforms state-of-the-art memory-efficient optimizers such as GaLore and Adam-Mini.

ACKNOWLEDGMENTS

This work used the Dutch national e-infrastructure with the support of the SURF Cooperative using the funding of the projects EINF-12538, EINF-10925 and NWO-2023.027. This work is partially supported by the SLAIDER project. We would like to express our deepest gratitude to the anonymous reviewers whose insightful comments and suggestions significantly improved the quality of this paper. Shiwei Liu is supported by the Royal Society with the Newton International Fellowship.

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

# A    STATISTICS ANALYSIS OF GRADIENT SPIKES ACROSS VARIOUS TYPES OF LAYERS

It is important to examine whether gradient spikes exhibit a preference for certain layers. To do so, we report the number of gradient spikes across various types of layers and the ratio of gradient spikes to the number of parameters in five types of layers: Embedding Layer, Attention Layer, FFN Layer, LayerNorm Layer, and LM_Head Layer. The experiments were conducted with LLaMA-60M on the C4 dataset, with gradient spikes detected over 1000 training steps. The detailed statistics are provided in Table 5. We observe the following: ❶ The Embedding Layer exhibits the highest number of gradient spikes, also it has the largest parameter count. ❷ The LayerNorm Layer, however, experiences an exceptionally high frequency of spikes, even with the smallest number of parameters.

Table 5: Number and Ratio of Gradient Spikes in each layer style of LLaMA. $\#Spikes$ are collected from 1000 training steps. Experiments are conducted with LLaMA-60M on C4.

| Module Name | #Total Spikes | #Total Params | $\frac{\#Total\ Spikes}{\#Total\ Params}$ |
|---|---|---|---|
| Embed | 11954001 | 16384000 | 0.729 |
| Attention | 86302 | 8388608 | 0.010 |
| FFN | 105415 | 16908288 | 0.006 |
| LayerNorm | 949302 | 8704 | 109.06 |
| LM_Head | 13893 | 16384000 | 0.000848 |

# B    LOCATIONS OF LOSS BUMPS AND GRADIENT SPIKES

To further investigate the correlation between loss bumps and gradient spikes, we present the locations of gradient spikes associated with the loss bumps in Table 6. The results reveal two key findings: ❶ Gradient spikes are presented in different layers associated with the loss bump; ❷ Gradient spikes typically occur before loss bumps, indicating that these gradient spikes may trigger loss bumps.

Table 6: Location of Spike Gradient at Each Layer for Different Tasks. The spike gradient is detected via GSS($g_i$) > 50. The experiments are based on LLaMA-60M and Pythia-70M.

| Model | Training Step When Loss Bump Occurs | Training Step When Spike Gradient Occurs in Each Layer | | | | | | | | | | | |
|---|---|---|---|---|---|---|---|---|---|---|---|---|---|
| | | 0th | 5th | 10th | 15th | 20th | 25th | 30th | 35th | 40th | 45th | 50th | 55th |
| LLaMA-60M (C4) | 198 | 202 | 196 197 198 | 197 | 197 205 278 | 196 197 198 202 | 196 197 198 199 | 197 198 | 197 205 | 197 198 201 205 | 197 198 | | 197 198 199 |
| LLaMA-60M (SlimPajama) | 207 328 394 | | 206 | 206 207 | 206 | 205 206 207 209 | 206 207 209 328 | 206 207 | 206 210 | 206 207 209 | 206 | 392 393 394 | 206 207 |
| Pythia-70M (C4) | 358 578 | | 571 577 578 | 573 577 | 357 571 577 578 | 357 358 574 576 577 578 | | | | | | | |

## C  PSEUDOCODE

---

**Algorithm 1:** `SPAM`

---

**Input:** A layer weight matrix $w \in \mathbb{R}^{m \times n}$, learning rate $\alpha$, decay rates $\beta_1 = 0.9, \beta_2 = 0.999$, initial parameters $w_0$, randomly initialize mask $\mathbf{M}$ with $d$ density for each layer, the first moment $m$, the second moment $v$, threshold $\theta$ for GSS, momentum rerest interval $\Delta T$, warmup scale total steps $N$, small constant $\epsilon = 1 \times 10^{-6}$. $T$ is total training steps.

**Output:** optimized parameters $w_T$.

**while** $t < T$ **do**

    Get $g_t \in \mathbb{R}^{m \times n} \leftarrow -\nabla_W \phi_t(w_t)$               ▷*Generate Gradients*

    $warmup\_scale = 1 - \texttt{CosineAnnealing}(Mod(t, \Delta T), N)$

    **if** $Mod\,(t, \Delta T) = 0$ **then**

        $\mathbf{M} \leftarrow \texttt{random.rand}(\theta.shape) < d$        ▷ *Random initialize the binary mask*

        $\mathbf{m} \leftarrow \texttt{zeros\_like}(\theta[\mathbf{M}])$           ▷ *reset the first moment to zero*

        $\mathbf{v} \leftarrow \texttt{zeros\_like}(\theta[\mathbf{M}])$          ▷ *reset the second moment to zero*

    $Spike\_M = g_t[\mathbf{M}] ** 2 > \theta * \mathbf{v}$            ▷ *Detect spiked gradients*

    **if** $sum(Spike\_M) > 0$ **then**

        $g_t[\mathbf{M}][Spike\_M] = \text{sign}(g_n[\mathbf{M}][Spike\_M]) \cdot \sqrt{\theta * \mathbf{v}[Spike\_M]}$    ▷ *Spike Gradients CLIP*

    $\mathbf{m}_t = \beta_1 \mathbf{m}_{t-1} + (1 - \beta_1)g_t$

    $\mathbf{v}_t = \beta_2 \mathbf{v}_{t-1} + (1 - \beta_2)g_t^2$

    $\hat{\mathbf{m}}_t = \frac{\mathbf{m}_t}{1 - \beta_1^t}$

    $\hat{\mathbf{v}}_t = \frac{\mathbf{v}_t}{1 - \beta_2^t}$

    $w_t = w_{t-1} - \alpha * warmup\_scale * \frac{\hat{\mathbf{m}}_t}{\sqrt{\hat{\mathbf{v}}_t} + \epsilon}$

    t=t+1

**Return:** optimized parameters $w_T$

---

## D  ARCHITECTURE AND HYPERPARAMETERS

We introduce details of the LLaMA architecture and hyperparameters used for pre-training, following Lialin et al. (2023a); Zhao et al. (2024). Table 7 shows the most hyperparameters of LLaMA models across model sizes. We use a max sequence length of 256 for all models, with a batch size of 512, with a batch size of 131K tokens. For all experiments, we adopt learning rate warmup of 1000 training steps, and use cosine annealing for the learning rate schedule, decaying to 10% of the initial learning rate.

Table 7: Configurations of LLaMA models used in this paper. Data amount are specified in #tokens.

| Params | Hidden | Intermediate | Heads | Layers | Steps | Data amount |
|--------|--------|--------------|-------|--------|-------|-------------|
| 60M    | 512    | 1376         | 8     | 8      | 10K   | 1.3B        |
| 130M   | 768    | 2048         | 12    | 12     | 20K   | 2.6B        |
| 350M   | 1024   | 2736         | 16    | 24     | 60K   | 7.8B        |
| 1 B    | 2048   | 5461         | 24    | 32     | 89K   | 11.6B       |

For all methods across each model size (from 60M to 1B), we tune the learning rates from $1e-4$ to $1e-3$ with an increasing step of $2 \times 10^{-4}$ for pre-training tasks, and the best learning rate is selected based on the validation perplexity. We find that the hyperparameters, Interval $\Delta T$ and warmup step $N$, are insensitive to model size and remain stable with the same learning rate across different model sizes. The detailed hyperparameter of `SPAM` on pre-training and fine-tuning are reported in Table 8 and Table 9.

Table 8: Hyperparameters of SPAM for pre-training experiments in this paper.

| Hyper-Parameters | LLaMA-60M | LLaMA-130M | LLaMA-350M | LLaMA-1B |
|---|---|---|---|---|
| Standard Pretraining | | | | |
| Learning rate | $1e-3$ | $8e-4$ | $4e-4$ | $2e-4$ |
| Interval $\Delta T$ | 500 | 500 | 500 | 500 |
| Threshold $\theta$ | 5000 | 5000 | 5000 | 5000 |
| Warmup steps $N$ | 150 | 150 | 150 | 150 |
| Memory-Efficient Pretraining | | | | |
| Learning rate | $4e-3$ | $4e-3$ | $2e-3$ | $5e-4$ |
| Interval $\Delta T$ | 500 | 500 | 500 | 1000 |
| Threshold $\theta$ | 5000 | 5000 | 5000 | 5000 |
| Warmup steps $N$ | 150 | 150 | 150 | 300 |

Table 9: Hyperparameters of SPAM for fine-tuning experiments in this paper.

| Hyper-Parameters | LLaMA2-7B |
|---|---|
| Standard Fine-tuning | |
| Learning rate | $5e-5$ |
| Interval $\Delta T$ | 1000 |
| Threshold $\theta$ | 5000 |
| Warmup steps $N$ | 300 |
| Memory-Efficient Fine-tuning | |
| Learning rate | $1e-4$ |
| Interval $\Delta T$ | 250 |
| Threshold $\theta$ | 5000 |
| Warmup steps $N$ | 5 |

# E    VISION TASKS

We further evaluate SPAM on vision task. Specifically, we conducted experiments on ImageNet-1K using ConvNeXt-Tiny (Liu et al., 2022b) and ViT-Tiny (Touvron et al., 2021). We adopt the default training recipe from the official code of ConvNeXT[2] and train all models for 120 epochs. We set $\Delta T = 25$K, $N = 20$ and $\theta = 5000$ for SPAM. The results in Table 10 demonstrate that SPAM can achieve on par or better performance than vanilla AdamW.

Table 10: SPAM performs on par or better than AdamW on vision tasks.

| Optimizer | Model | Metric | 25% steps | 50% steps | 75% steps | 100% steps |
|---|---|---|---|---|---|---|
| AdamW | ConNeXt-T | Test Acc ($\uparrow$) | 68.15 | 74.00 | 78.83 | 80.89 |
| SPAM | ConNeXt-T | Test Acc ($\uparrow$) | 68.36 | 73.63 | 78.85 | 81.04 |
| AdamW | ViT-Tiny | Test Acc ($\uparrow$) | 48.09 | 56.93 | 65.06 | 69.71 |
| SPAM | ViT-Tiny | Test Acc ($\uparrow$) | 47.34 | 56.47 | 65.57 | 69.98 |

---

[2]https://github.com/facebookresearch/ConvNeXt

## F  MORE ABLATION STUDY OF SUBSET SELECTIOIN STRATEGIES

Key questions surrounding sparse momentum include how to effectively select parameter subset and whether to retain momentum for weights that are sampled multiple times. To answer this questions, we conduct comparative studies based on LLaMA-60M and C4 and the results are shown in Figure 7. Figure 7-Left shows the performance of three subset selection strategies where we will reset all moments after each momentum reset and keep gradients for all unselected parameters. Figure 7-Middle shows the performance of three subset selection strategies where we will keep the overlapped moments after each momentum reset and keep gradients for all unselected parameters. Figure 7-Right shows the performance of three subset selection strategies where we will reset all the moments after each momentum reset and drop gradients for all unselected parameters in each updating step. We observe the following: ❶ Among the three subset selection strategies—Max weight magnitude-based, Max gradient magnitude-based, and Random selection—the Random selection consistently outperforms the other two approaches. ❷ Comparing Figure 7-Left and Figure 7-Right, we see that resetting all moments after each momentum reset yields better performance than preserving overlapping moments.

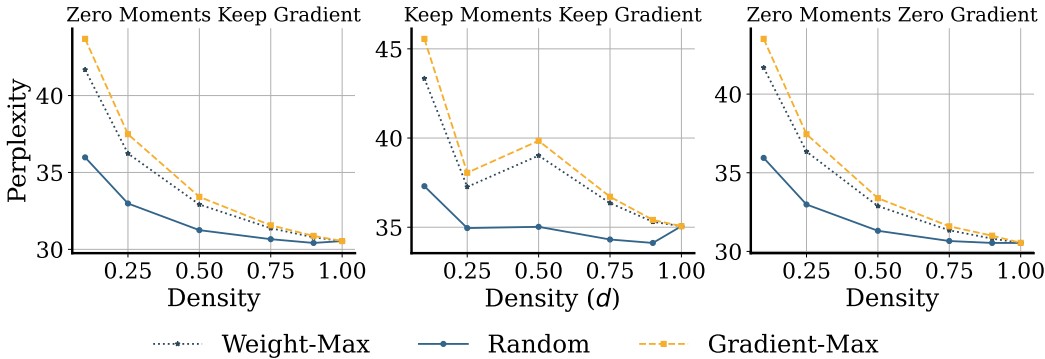

Figure 7: Ablations for subset selection strategies. The experiments are conducted with LLaMA-60M on C4.

## G  EXPERIMENTS ON TIME SERIES DATA

To showcase SPAM's ability to mitigate gradient spikes across a broader range of applications, we conducted additional experiments on time-series prediction tasks. In these experiments, we intentionally introduced anomalous data with a 10% probability to simulate gradient anomalies. Experiments are conducted with 10 repeated runs on Weather time series data[3] using PatchTST (Nie et al., 2023) model. The results are presented in Figure 8

The findings demonstrate that as the severity of anomalous data increases, SPAM's performance advantage over Adam becomes more pronounced, highlighting its effectiveness in mitigating the adverse impact of gradient spikes.

## H  PROLONGED DETRIMENTAL EFFECTS OF GRADIENT SPIKES DURING REAL TRAINING

We also measure the values of gradient, first moment, and second moment during the training of LLaMA-60M on the C4 dataset. The results are now presented in Figure 9.

From the figure, we observe that during actual training, gradient spikes also have a significant and prolonged detrimental impact on moments, especially on the second moment, providing further evidence to support our claims.

---

[3]https://www.bgc-jena.mpg.de/wetter/

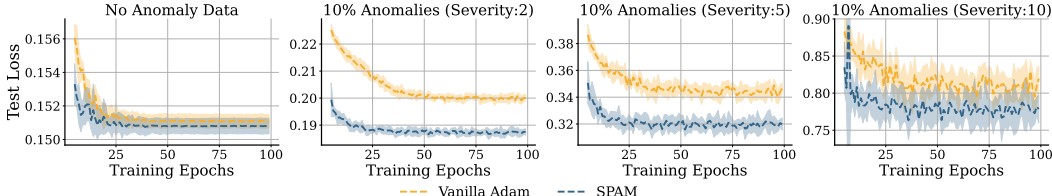

Figure 8: **Test Loss during Training Process on Weather Time-series Data.** Anomalous data is generated by adding Gaussian noise to 10% of randomly selected input values. Specifically, the anomalies data are conducted with $X = X + \texttt{Gaussin}(0, \texttt{Severity} * \texttt{Max}(X))$ where $X$ is the inputs.

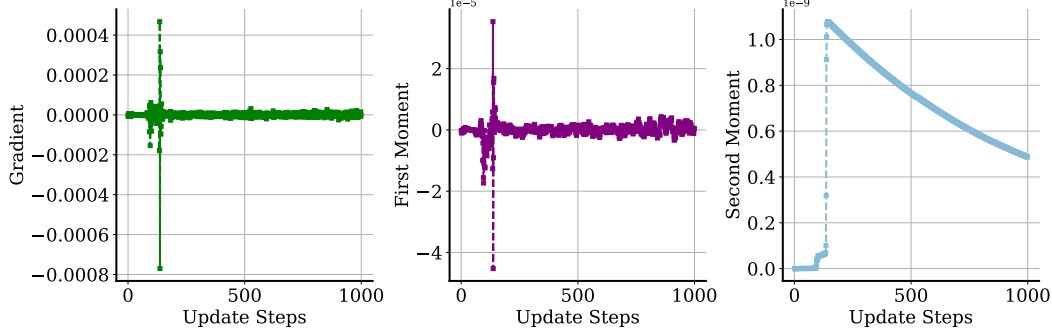

Figure 9: **Gradient spikes have prolonged detrimental effects on the first and second moments.** Experiments are conducted on C4 dataset with LLaMA-60M.

## I  SENSITIVITY ANALYSIS OF HYPERPARAMETER $\theta$ ON LLM ARCHITECTURES

We conducted experiments to evaluate the sensitivity of the gradient spike clipping threshold, $\theta$, across three widely used LLM architectures: LLaMA, Pythia, and OPT. These experiments were performed on pre-training tasks using the C4 dataset. The final perplexity is reported in Table 11.

The results indicate that the gradient spike clipping threshold is not highly sensitive to the choice of LLM architecture. SPAM consistently outperforms Adam across a wide range of $\theta$. Furthermore, the optimal range for $\theta$ lies between 1000 and 5000.

Table 11: Sensitivity Analysis of Hyperparameter $\theta$ on LLM architectures. Perplexity is reported.

| Architectures | $\theta = 500$ | $\theta = 1000$ | $\theta = 2500$ | $\theta = 5000$ | $\theta = 10000$ | Adam |
|---|---|---|---|---|---|---|
| LLaMA-60M | 30.77 | 30.59 | 30.57 | 30.46 | 30.82 | 34.09 |
| Pythia-70M | 34.4 | 34.1 | 34.1 | 34.2 | 35.1 | 38.34 |
| OPT-125M | 28.7 | 28.4 | 28.5 | 28.6 | 29.0 | 32.20 |

## J  GSS VS. DISTRIBUTION BASED CLIPPING

We conducted an experiment using an outlier detection mechanism based on the assumption that stochastic gradient distributions follow a Gaussian distribution, as suggested in (Simsekli et al., 2019; Chaudhari & Soatto, 2018; Mandt et al., 2016):

$$G_{batch} \sim \mathcal{N}(G, \delta^2 \mathbf{I}),$$

where $G_{batch}$ is the stochastic gradient, $G$ represents the gradient over the entire dataset, and $\delta^2$ is the variance. Since calculating $G$ on-the-fly during training is computationally infeasible, we approximate it using the moving average of $G_{batch}$. The variance $\delta^2$ is estimated online as:

$\delta^2 = \frac{1}{N} \sum_{n=1}^{N} \left( G_{batch}^{(n)} - G^{(n)} \right)^2$, where $N$ is the total training steps. Gradients are then evaluated element-wise, and any element $G_{batch}^{(n)}$ satisfying: $|G_{batch}^{(n)} - G^{(n)}| > 3\delta$ is identified as an outlier. Such outlier elements are clipped to satisfy: $|G_{batch}^{(n)} - G^{(n)}| = 3\delta$.

We conducted experiments using LLaMA-60M and LLaMA-130M to evaluate the performance of this Gaussian-based Clipping and compare it with our proposed GSS-based clipping. The results are reported in Table 12. As the table indicates, Gaussian-based clipping falls short of our GSS-based clipping. One possible explanation is that stochastic gradient distributions are very complex and Gaussian distribution can not reflect the true distribution.

Table 12: Comparison between SPAM with spike-aware clipping and Gaussian-based clipping.

| Methods | LLaMA-60M | LLaMA-130M |
|---|---|---|
| SPAM w/GSS based clipping | 30.46 | 23.36 |
| SPAM w/ Gaussian based Clipping | 30.83 | 25.93 |

## K   GSS BASED CLIPPING VS. NULLIFYING

We conducted experiments on LLaMA-60M and LLaMA-130M to compare the performance of Spike-Aware Clipping and Nullifying Gradient Spikes. As shown in Table 13 and Table 14, SPAM with Spike-Aware Clipping outperforms SPAM with Nullifying on both pre-training and fine-tuning tasks, demonstrating the effectiveness of Spike-Aware Clipping.

Table 13: Comparison between SPAM w/ spike-aware clipping and SPAM w/ nullifying gradient spikes.

| Methods | LLaMA-60M | LLaMA-130M |
|---|---|---|
| SPAM w/ Spike Aware Clipping | 30.46 | 23.36 |
| SPAM w/ Nullifying | 30.86 | 23.62 |

Table 14: Comparison between SPAM w/ spike-aware clipping and SPAM w/ nullifying gradient spikes on fine-tuning task. The experiments are based on a pre-trained OPT-1.3B model.

| Methods | WinoGrande | COPA |
|---|---|---|
| SPAM w/ Spike Aware Clipping (d=100%) | 59.4 | 79.0 |
| SPAM w/ Spike Aware Clipping (d=0.25%) | 58.3 | 75.0 |
| SPAM w/ Nullifying(d=100%) | 58.0 | 78.0 |
| SPAM w/ Nullifying (d=0.25%) | 57.4 | 75.0 |

## L   COMPUTATIONAL ANALYSIS

We measured the running time per iteration for both LLaMA-60M and LLaMA-130M. The results, presented in Table 15, indicate that SPAM incurs a slightly higher computational overhead compared to Adam, Adam-mini, and Adafactor. This overhead is primarily due to the gradient spike detection operation and the gradient selection based on sparse masks. However, we believe that such a small overhead is negligible compared to the overall pre-training time which can be dozens or hundreds of hours.

Table 15: Running Time per Iteration (second). The runtime is measured by the average of 100 iterations under one H100 GPU.

| Method | Time per Iteration (LLaMA-60M) | Time per Iteration (LLaMA-130M) |
|---|---|---|
| Adam | 0.3666 (s) | 0.6397 (s) |
| Adam-mini | 0.3614 (s) | 0.6472 (s) |
| Adafactor | 0.3778 (s) | 0.6565 (s) |
| GaLore (rank=128) | 0.3871 (s) | 0.6702 (s) |
| SPAM(d=100%) | 0.3814 (s) | 0.6683 (s) |
| SPAM(d=25%) | 0.3799 (s) | 0.6658 (s) |

