# OpenReview forum: "SPAM: Spike-Aware Adam with Momentum Reset for Stable LLM Training"
_ICLR.cc/2025/Conference — ICLR 2025 Poster_

### Official Review · Reviewer_BU76 · 2024-11-03

**Soundness:** 3
**Presentation:** 3
**Contribution:** 3
**Rating:** 6
**Confidence:** 4

**Summary:**

This paper analyzes the phenomenon of loss spikes observed in large language models (LLMs) and reveals that loss spikes are not restricted to specific layers or architectures, but occur in a wide range of environments. The authors experimentally demonstrate that loss spikes affect the performance of AI systems and mathematically show that loss spikes are influenced by momentum-based optimizers, such as Adam. The proposed method, SPAM (Stochastic Gradient Projection with Adaptive Momentum), effectively addresses this issue by using a threshold-based approach to manage the average gradient. The paper compares the performance of SPAM in both pre-training and fine-tuning stages with various other methods, showing its effectiveness.

**Strengths:**

This paper addresses the loss spike problem from the perspective of gradient clipping and demonstrates the algorithm's validity through an ablation study on the hyper-parameters used in the algorithm, along with various performance improvements. Additionally, the paper proposes a memory-efficient algorithm using sparse momentum, aiming to solve both the loss spike issue and the out-of-memory problem simultaneously.

**Weaknesses:**

1. Clipping gradients based on a threshold seems to lack novelty. It might be worthwhile to consider methods that prevent gradient spikes altogether.
2. In sparse momentum, a random mask is applied, setting certain gradients to zero. It would be helpful to explain in detail how this actually reduces memory usage. From an algorithmic perspective, it appears as though the entire matrix, including the zero elements, is still being stored.

**Questions:**

1. The use of theta in GSS seems heuristic. How about selecting outliers based on the known distribution (such as Gaussian) of gradients instead? [Umut Simsekli, Levent Sagun, & Mert Gurbuzbalaban. (2019). A Tail-Index Analysis of Stochastic Gradient Noise in Deep Neural Networks.]
2. If the method used in the experiments is SPAM with the sparse momentum approach, according to Algorithm 1, when m and v are set to zero, some weights are not updated, similar to dropout. It’s unclear whether the performance improvement is due to this or the actual spike gradient clipping.

---

> ### Author Response · Authors · 2024-11-21
> **Response to Reviewer BU76 (1/2)**
>
> We sincerely appreciate your detailed comments and positive ranking. We address your questions below.
>
> **Comment 1: Clipping gradients based on a threshold seems to lack novelty. It might be worthwhile to consider methods that prevent gradient spikes altogether.**
>
> - Thank you for comment! Following your suggestion, we integrated SPAM with a spike-prevening technique, Scaled Embed proposed in [1]. Scaled Embed was introduced to prevent gradient spikes by scaling embeddings by $\sqrt{d}$, where $d$ denotes the hidden size of the transformer.
>
> - Experiments were conducted using LLaMA-60M on the C4 dataset. The results, presented in the table below, demonstrate that Scaled Embed alone can slightly reduce perplexity. The combination of Scaled Embed and SPAM further reduces the final perplexity.
>
> - Moreover, we would like to emphasize that, to the best of our knowledge, we are the first to propose a threshold-based gradient clipping method to effectively mitigate the negative impact of gradient spikes. While previous work [1] was proposed to prevent gradient spikes, it primarily focuses on modifying the initialization. In contrast, our approach tackles this issue from the perspective of optimization, which is novel.
>
>     **Table: The performance of Combining SPAM and Scaled Embed (from [1]).**
>
>     | Methods            | LLaMA-60M |
>     |---------------------|-----------|
>     | Adam               | 34.09     |
>     | SPAM               | 30.46     |
>     | Scaled Embed       | 33.87     |
>     | SPAM + Scaled Embed| 30.36     |
>
> [1] Takase, Sho, et al. "Spike No More: Stabilizing the Pre-training of Large Language Models." arXiv preprint arXiv:2312.16903 (2023).
>
> **Comment 2: In sparse momentum, a random mask is applied, setting certain gradients to zero. It would be helpful to explain in detail how this actually reduces memory usage. From an algorithmic perspective, it appears as though the entire matrix, including the zero elements, is still being stored.**
>
> - SPAM reduces training memory usage by storing only the selected elements (mask=1) of the first and second moments into vectors, rather than sparse matrices with zero elements. Specifically, when we apply a random mask with a density $d=0.25$, only 25\% of the gradient elements are retained and used to update the first and second moment vectors. This means that SPAM initializes and maintains the first and second moment vectors that are only 25\% the size of the corresponding weight matrices.
>
> - We provide a pytorch code for our initialization of first and second moments in SPAM to aid understanding.
> ```bash
> grad=p.grad[Mask]
> # Mask is the random selected sparse mask, 'p.grad[Mask]' extracts a vector of gradients only where the mask is True.
>
> state["exp_avg"] = torch.zeros_like(grad)
> # This initializes the first moment (exponential moving average of gradients) with the same shape as 'grad',
> # which is significantly reduced compared to the original gradient size, being %d of the original size.
>
> state["exp_avg_sq"] = torch.zeros_like(grad)
> # Similarly, this initializes the second moment (exponential moving average of squared gradients) with the same shape as 'grad',
> ```

---

> ### Author Response · Authors · 2024-11-21
> **Response to Reviewer BU76 (2/2)**
>
> **Comment 3:  The use of theta in GSS seems heuristic. How about selecting outliers based on the known distribution (such as Gaussian) of gradients instead? [Umut Simsekli, Levent Sagun, \& Mert Gurbuzbalaban. (2019). A Tail-Index Analysis of Stochastic Gradient Noise in Deep Neural Networks.**
>
>
> - Thank you for your insightful suggestion. Following your recommendation, we conducted an experiment using an outlier detection mechanism based on the assumption that stochastic gradient distributions follow a Gaussian distribution, as suggested in [1, 2, 3]: $G_{\text{batch}} \sim \mathcal{N}(G, \delta^2 \mathbf{I}),$ where \( $G_{\text{batch}}$ \) is the stochastic gradient, \( G \) represents the gradient over the entire dataset, and \( $\delta^2$ \) is the variance. Since calculating \( G \) on-the-fly during training is computationally infeasible, we approximate it using the moving average of \( $G_{\text{batch}}$ \). The variance \( $\delta^2$ \) is estimated online as: $
> \delta^2 = \frac{1}{N} \sum_{n=1}^{N} \left( G_{\text{batch}}^{(n)} - G^{(n)} \right)^2,$ where \( N \) is the total training steps. Gradients are then evaluated element-wise, and any element \( $G_{\text{batch}}^{(n)}$ \) satisfying: $
> \left| G_{\text{batch}}^{(n)} - G^{(n)} \right| > 3\delta $ is identified as an outlier. Such outlier elements are clipped to satisfy:$ \left| G_{\text{batch}}^{(n)} - G^{(n)} \right| = 3\delta. $
>
> - We conducted experiments using LLaMA-60M and LLaMA-130M to evaluate the performance of this Gaussian-based clipping and compare it with our proposed GSS-based clipping. The results are reported in the table below. As the table indicates, Gaussian-based clipping falls short of our GSS-based clipping. One possible explanation is that stochastic gradient distributions are very complex and the Gaussian distribution cannot reflect the true distribution.
>
> - We appreciate your suggestion, as it allowed us to explore alternative approaches and strengthen the validation of our method. We will include a discussion of these findings in the revised version to provide a more comprehensive understanding of the effectiveness of our spike-aware clipping technique.
>
>
>     **Table: Comparison between SPAM with spike-aware clipping and Gaussian-based clipping.**
>
>     | Methods                        | LLaMA-60M | LLaMA-130M |
>     |--------------------------------|-----------|------------|
>     | SPAM w/ GSS-based clipping  (ours)   | 30.46     | 23.36      |
>     | SPAM w/ Gaussian-based clipping| 30.83     | 25.93      |
>
> [1] Umut Simsekli, Levent Sagun, \& Mert Gurbuzbalaban. (2019). A Tail-Index Analysis of Stochastic Gradient Noise in Deep Neural Networks.
>
> [2] Chaudhari, Pratik, and Stefano Soatto. "Stochastic gradient descent performs variational inference, converges to limit cycles for deep networks." 2018 Information Theory and Applications Workshop (ITA). IEEE, 2018.
>
> [3] Mandt, Stephan, Matthew Hoffman, and David Blei. "A variational analysis of stochastic gradient algorithms." International conference on machine learning. PMLR, 2016.
>
> **Comment 4: If the method used in the experiments is SPAM with the sparse momentum approach, according to Algorithm 1, when m and v are set to zero, some weights are not updated, similar to dropout. It’s unclear whether the performance improvement is due to this or the actual spike gradient clipping.**
>
> - Thank you for your insightful comment! We appreciate your observation regarding the similarities between the sparse momentum reset in SPAM and dropout. However, there are key distinctions between the two: **(1)**  In SPAM, the random sparse mask is updated only after every $\Delta T$ steps of training, whereas dropout typically applies random masking at every forward pass. **(2)** In SPAM, during each sparse momentum reset, both the first moment (M) and second moment (V) are reset to zero. In contrast, dropout does not reset cached moments; it merely selects a random subset of moments for weight updates.
>
> - To address your concern, we conducted additional experiments applying dropout with probabilities p = 0.25 and p = 0.5 to the moments. The results, presented in the table below, indicate that dropout has an adverse impact on performance. This further validates that the superior performance of SPAM is not due to a dropout-like effect.
>
>     **Table: The performance of moment dropout.**
>
>     | Methods                            | LLaMA-60M |
>     |------------------------------------|-----------|
>     | Adam                               | 34.09     |
>     | Adam w/ Moment dropout (p=0.5)     | 35.39     |
>     | Adam w/ Moment dropout (p=0.25)    | 38.23     |
>     | SPAM                               | 30.46     |

---

> > ### Author Response · Authors · 2024-11-27
> >
> > Dear Reviewer BU76,
> >
> > We sincerely thank you for your positive rating and thoughtful feedback on our paper. We deeply appreciate the time and effort you have dedicated to reviewing our work and providing such valuable insights.
> >
> > As the rebuttal period is still ongoing, we wanted to kindly check if you have any additional questions or concerns that we can address. Your insights are incredibly important to us, and we are more than happy to provide further clarifications or information if needed.
> >
> > Thank you once again for your time, thoughtful consideration, and support.
> >
> >
> > Kind regards,
> >
> > The Authors

---

### Official Review · Reviewer_LnTM · 2024-11-04

**Soundness:** 3
**Presentation:** 3
**Contribution:** 3
**Rating:** 6
**Confidence:** 4

**Summary:**

The paper introduces SPAM (Spike-Aware Adam with Momentum Reset), an innovative optimization approach aimed at stabilizing the training of LLMs. SPAM addresses the challenges of gradient and loss spikes that result in training instability, which can require costly interventions such as checkpoint recovery. The proposed method improves the ADAM optimizer via integrating two key mechanisms:  spike-aware gradient clipping and momentum reset. These novelties contribute to mitigating the accumulation of gradient spikes, therefore enhancing the training stability and efficiency. Extensive experiments show that SPAM outperforms ADAM and other memory-efficient optimizers like GaLore and Adam-Mini over various LLM sizes and tasks, supporting its potential to improve training under memory constraints.

**Strengths:**

1. The integration of momentum reset and spike-aware gradient clipping into Adam is noval and addresses the persistent issue of gradient spikes in Large Language Model training.
2. The experiments are thorough and extensive, with evaluations spanning multiple LLM architectures and scales. The results clearly manifest SPAM's superior performance over the standard and memory-efficient baselines.
3. The approach is highly relevant, especially for large-scale training where stability and efficiency are paramount.
4. SPAM's sparse momentum feature is especially useful for resource-constrained training, making it an important contribution to memory-efficient optimization approaches.

**Weaknesses:**

1. The paper mentions the efficient implementation of momentum reset and spike detection, but a moredetailed practical guidance or pseudo code might improve reproducibility.
2. While SPAM performs excellently across various Large Language Model sizes, additional experiments on tasks beyond LLM training, such as CV models or multi-task learning, should illustrate broader applicability.
3. The choice of the gradient spike threshold might affect performance to a great extent. More discussion on how to tune this parameter across different model architectures would be of benefits.

**Questions:**

Could the spike-aware clipping method proposed be extended to handle other types of optimization tasks and scenarios, like adversarial training?

---

> ### Author Response · Authors · 2024-11-21
> **Response to Reviewer LnTM (1/2)**
>
> We are grateful for your support and detailed comments! We provide responses to address your comments below.
>
> **Comment 1: The paper mentions the efficient implementation of momentum reset and spike detection, but a moredetailed practical guidance or pseudo code might improve reproducibility.**
>
> - Thank you for your comment. We have included the pseudocode for SPAM in our submission. Additionally, we have submitted the full code along with detailed README in the Supplementary Material. For your reference, the pseudocode is located in **Appendix C**.
>
> **Comment 2: While SPAM performs excellently across various Large Language Model sizes, additional experiments on tasks beyond LLM training, such as CV models or multi-task learning, should illustrate broader applicability.**
>
> - Thank you for your thoughtful comment. In this work, our primary focus is on mitigating the impact of gradient spikes in LLM training, as indicated in the title. However, to demonstrate the broader applicability of SPAM, we also evaluated its performance on CV models, as detailed in **Appendix E**. For your convenience, we have summarized the results in the table below. These results show that SPAM achieves performance comparable to or better than vanilla AdamW. The lack of significant improvement on ImageNet may be attributed to the dataset’s inherently clean nature.
>
>     **Table: SPAM Performs on Par or Better than AdamW on Vision Tasks**
>
>     | Optimizer | Model      | Metric               | Final Checkpoint |
>     |-----------|------------|----------------------|------------------|
>     | AdamW     | ConNeXt-T  | Test Acc ($\uparrow$) | 80.89           |
>     | SPAM      | ConNeXt-T  | Test Acc ($\uparrow$) | 81.04           |
>     | AdamW     | ViT-Tiny   | Test Acc ($\uparrow$) | 69.71           |
>     | SPAM      | ViT-Tiny   | Test Acc ($\uparrow$) | 69.98           |
>
> - Furthermore, to showcase SPAM's ability to mitigate gradient spikes across a broader range of applications, we conducted additional experiments on time-series prediction tasks. In these experiments, we intentionally introduced anomalous data with a 10\% probability to simulate gradient anomalies. The results are presented in the table below.
>
> - The findings demonstrate that as the severity of anomalous data increases, SPAM's performance advantage over Adam becomes more pronounced, highlighting its effectiveness in mitigating the adverse impact of gradient spikes.
>
>     **Table: Performance on Weather Time-Series Data.** Anomalous data is generated by adding Gaussian noise to 10% of randomly selected input values. Specifically, anomalies are created using \( X = X + $\texttt{Gaussian}$(0, $\texttt{Severity}$ * $\texttt{Max}$(X)) \), where \( X \) represents the inputs. Mean and standard deviation of MSE ($\downarrow$) across 10 repeated runs are reported.
>
>     | Optimizer | W/o Anomalies       | W/ Anomalies (severity=2) | W/ Anomalies (severity=5) | W/ Anomalies (severity=10) |
>     |-----------|---------------------|---------------------------|---------------------------|----------------------------|
>     | AdamW     | 0.151 ± 0.004       | 0.200 ± 0.001            | 0.346 ± 0.004            | 0.819 ± 0.016             |
>     | SPAM      | 0.1505 ± 0.004      | 0.187 ± 0.001            | 0.319 ± 0.004            | 0.778 ± 0.018             |
>
>
> **Comment 3: The choice of the gradient spike threshold might affect performance to a great extent. More discussion on how to tune this parameter across different model architectures would be of benefits.**
>
> - Thank you for your valuable comment. Following your suggestion, we conducted experiments to evaluate the sensitivity of the gradient spike clipping threshold, $\theta$, across three widely used LLM architectures: LLaMA, Pythia, and OPT.  These experiments were performed on pre-training tasks using the C4 dataset. The final perplexity is reported in the below table.
>
> - The results indicate that the gradient spike clipping threshold is not highly sensitive to the choice of LLM architecture. SPAM consistently outperforms Adam across a wide range of $\theta$. Furthermore, the optimal range for $\theta$ lies between 1000 and 5000. We also include this analysis in Appendix I.
>
>     **Table: Sensitivity Analysis of Hyperparameter θ on LLM architectures. Perplexity is reported.**
>
>     | Architectures  | θ=500  | θ=1000 | θ=2500 | θ=5000 | θ=10000 | Adam   |
>     |----------------|---------|--------|--------|--------|---------|--------|
>     | LLaMA-60M      | 30.77   | 30.59  | 30.57  | 30.46  | 30.82   | 34.09  |
>     | Pythia-70M     | 34.4    | 34.1   | 34.1   | 34.2   | 35.1    | 38.34  |
>     | OPT-125M       | 28.7    | 28.4   | 28.5   | 28.6   | 29.0    | 32.20  |

---

> > ### Author Response · Authors · 2024-11-21
> > **Response to Reviewer LnTM (2/2)**
> >
> > **Comment 4: Could the spike-aware clipping method proposed be extended to handle other types of optimization tasks and scenarios, like adversarial training?**
> >
> > -  Thank you for your insightful comment! We agree that the spike-aware clipping method, similar to traditional gradient clipping techniques, has the potential to be extended to various optimization tasks, including adversarial training.
> >
> > - Following your suggestion, we conducted a preliminary evaluation on PGD-based adversarial training using the code from [https://github.com/locuslab/robust_overfitting](https://github.com/locuslab/robust_overfitting). We measured both natural accuracy and robust accuracy under PGD-10 attacks at the best checkpoint and the final checkpoint. Our results indicate a slight improvement in both metrics when applying the spike-aware clipping method.
> >
> > - However, drawing a solid conclusion would require more comprehensive evaluations, which are beyond the scope of this work. We acknowledge this as an exciting direction for future research and appreciate your suggestion.
> >
> >     **Table: The spike-aware clipping for PGD-Adversarial training. Experiments are based on ResNet18 and CIFAR10. Threshold θ is set to 1000.**
> >
> >     | Methods                     | Metrics      | Final Checkpoint | Best Checkpoint |
> >     |-----------------------------|--------------|------------------|-----------------|
> >     | W/ the spike-aware clipping | Natural ACC  | 84.3             | 82.2           |
> >     | W/ the spike-aware clipping | Robust ACC   | 45.9             | 53.1           |
> >     | W/o the spike-aware clipping| Natural ACC  | 83.8             | 82.2           |
> >     | W/o the spike-aware clipping| Robust ACC   | 45.6             | 53.0           |

---

> > > ### Author Response · Authors · 2024-11-27
> > >
> > > Dear Reviewer LnTM,
> > >
> > > We sincerely thank you for your positive rating and thoughtful feedback on our paper. We deeply appreciate the time and effort you have dedicated to reviewing our work and providing such valuable insights.
> > >
> > > As the rebuttal period is still ongoing, we wanted to kindly check if you have any additional questions or concerns that we can address. Your insights are incredibly important to us, and we are more than happy to provide further clarifications or information if needed.
> > >
> > > Thank you once again for your time, thoughtful consideration, and support.
> > >
> > >
> > > Kind regards,
> > >
> > > The Authors

---

> > > > ### Comment · Reviewer_LnTM · 2024-11-28
> > > > **Thanks for the response**
> > > >
> > > > Thanks for the authors' response which solves most of my concerns. After careful consideration, I decide to maintain my score.

---

> > > > > ### Author Response · Authors · 2024-11-28
> > > > > **Thank you**
> > > > >
> > > > > We sincerely appreciate your thorough review and valuable feedback. We are pleased to have successfully addressed your concerns, and your suggestions have significantly improved the quality of our manuscript. Thank you for your time, effort, and dedication throughout this review process.
> > > > >
> > > > > Best regards,
> > > > >
> > > > > the Authors

---

### Official Review · Reviewer_BUJ8 · 2024-11-04

**Soundness:** 2
**Presentation:** 2
**Contribution:** 3
**Rating:** 6
**Confidence:** 4

**Summary:**

The paper proposes a novel optimization method designed to mitigate training instabilities, specifically gradient and loss spikes in large language models (LLMs). The method, SPAM, introduces momentum reset and spike-aware gradient clipping to counteract the effects of significant gradient spikes that can disrupt the learning process. Extensive experiments suggest that SPAM outperforms traditional Adam and its memory-efficient variants, offering better stability and performance across different model scales and training setups.

**Strengths:**

- The paper’s analysis highlighting the prevalence and impact of gradient spikes in LLM training is insightful and demonstrates an important issue.
- The experiments show consistent improvements across different LLM sizes and benchmarks, suggesting the method’s robustness within these settings.
- The introduction of sparse momentum is a useful addition for reducing the memory overhead of training large models.

**Weaknesses:**

- Although SPAM is compared to Adam and a few memory-efficient optimizers, it lacks comprehensive analysis against more recent memory-efficient methods. Furthermore, additional experiments, as outlined below, are necessary to strengthen the evaluation.

**Questions:**

1. **Detrimental effects of gradient spikes**. It would be valuable to observe the middle and right plots of Figure 5 during actual training.
2. **Moment reset.** Does the benefit of momentum reset lie in isolating the effects of gradient spikes, even at the cost of training intervals affected by these spikes?
3. **Statistical significance**. Were the fine-tuning experiments conducted using multiple random seeds?
4. **Baselines**. How does this method compare with other memory-efficient approaches such as MeZO [1], SparseMeZO [2], and Extremely Sparse MeZO [3]?
5. **Ablation studies**.
    1. There is no comparison between Spike-Aware Clipping and simply nullifying gradient spikes.
    2. It would be interesting to see if the parameter for sparse momentum could be selected in a structured manner, as described in  [4].
    3. What happens in the case $\triangle T < N$?
6. **Loss and Gradient plots of SPAM**. How do the loss and gradient plots of SPAM compare with those of other methods shown in Figures 2-4?
7. **Computational analysis**. Can the authors report the computational overhead of SPAM compared to other methods?

[1] Malladi, Sadhika, et al. "Fine-tuning language models with just forward passes." *Advances in Neural Information Processing Systems* 36 (2023): 53038-53075.

[2] Liu, Yong, et al. "Sparse mezo: Less parameters for better performance in zeroth-order llm fine-tuning." *arXiv preprint arXiv:2402.15751* (2024).

[3] Guo, Wentao, et al. "Zeroth-Order Fine-Tuning of LLMs with Extreme Sparsity." *arXiv preprint arXiv:2406.02913* (2024).

[4] He, Yang, and Lingao Xiao. "Structured pruning for deep convolutional neural networks: A survey." IEEE transactions on pattern analysis and machine intelligence (2023).

---

> ### Author Response · Authors · 2024-11-21
> **Response to Reviewer BUJ8 (1/3)**
>
> We sincerely appreciate your detailed comments. We provide point-wise responses to address your concerns below.
>
> **Comment 1: Although SPAM is compared to Adam and a few memory-efficient optimizers, it lacks comprehensive analysis against more recent memory-efficient methods. Furthermore, additional experiments, as outlined below, are necessary to strengthen the evaluation.**
>
> - Thank you for your helpful feedbacks. We conducted all the experiments you mentioned and presented below.
>
> **Comment 2: Detrimental effects of gradient spikes. It would be valuable to observe the middle and right plots of Figure 5 during actual training.**
>
> - Thank you for your valuable suggestions. Based on your feedback, we also measure the values of gradient, first moment, and second moment during the training of LLaMA-60M on the C4 dataset. The results are now presented in Figure 9 of Appendix H in our revision.
>
> - From the figure, we observe that during actual training, gradient spikes also have a significant and prolonged detrimental impact on moments, especially on the second moment, providing further evidence to support our claims.
>
> **Comment 3: Moment reset. Does the benefit of momentum reset lie in isolating the effects of gradient spikes, even at the cost of training intervals affected by these spikes?**
>
>
> - If we have understood your question correctly (please correct us if we misunderstand your question), you are asking whether the benefit of Momentum Reset lies in mitigating the effects of gradient spikes, even at the cost of losing accumulated momentum information. Our answer is yes.
>
> - The success of **SPAM** demonstrates that, instead of mindlessly accumulating Adam's moments with potentially spiked gradients (as gradient spikes occasionally occur), it is more effective to periodically reset the momentum to counteract their negative effects. However, this does not imply that momentum is unimportant for SPAM. On the contrary, we reset momentum only infrequently—every $\Delta T = 500$ steps—allowing SPAM to accumulate sufficient momentum during most of the training process.
>
> - This insight is further validated by the results presented in the table below. Periodically resetting momentum improves performance compared to Adam. However, the performance gains diminish when $\Delta T$ becomes too small, indicating the importance of balancing reset frequency with momentum accumulation.
>
>     **Table: The performance of decreasing the momentum reset interval ΔT (Experiments are based on LLaMA-130M and C4)**
>     |                  | Adam  | ΔT = 2500 | ΔT = 1000 | ΔT = 500 | ΔT = 250 | ΔT = 100 | ΔT = 50 |
>     |------------------|-------|-----------|-----------|----------|----------|----------|---------|
>     | **Perplexity**   | 24.91 | 23.72     | 23.45     | 23.36    | 23.60    | 24.28    | 27.06   |

---

> ### Author Response · Authors · 2024-11-21
> **Response to Reviewer BUJ8 (2/3)**
>
> **Comment 4: Statistical significance. Were the fine-tuning experiments conducted using multiple random seeds?**
>
> - Thank you for your question. The fine-tuning results were originally averaged over three runs. To address your concern more comprehensively, we have now reported the mean and standard deviation based on 10 runs with different random seeds. The updated table clearly illustrates that SPAM significantly outperforms other memory-efficient methods.
>
>     **Table: Fine-tuning performance of LLaMa$2$-$7$B on various downstream tasks**
>     The "Mem." column denotes the running GPU memory. The mean and standard deviation of 10 repeated experiments are reported.
>
>     | **Method**             | **Mem. (GB)** | **BoolQ**            | **PIQA**            | **SIQA**            | **HellaSwag**       | **WinoGrande**      | **ARC-e**          | **ARC-c**          | **OBQA**           | **Avg.**           |
>     |-------------------------|---------------|-----------------------|----------------------|----------------------|----------------------|----------------------|---------------------|---------------------|---------------------|---------------------|
>     | **Adam (Full FT)**      | 61G           | 79.7 ± 0.1           | 79.1 ± 0.1          | 51.3 ± 0.05         | 58.5 ± 0.02         | 74.8 ± 0.2          | **79.2 ± 0.1**     | 48.2 ± 0.01        | 36.2 ± 0.2         | 63.4 ± 0.1         |
>     | **LoRA**               | 26G           | 75.8 ± 0.4           | 79.0 ± 0.1          | 56.3 ± 0.1          | **59.9 ± 0.04**     | 79.6 ± 0.2          | 77.6 ± 0.1         | 46.9 ± 0.1         | 34.4 ± 0.3         | 63.7 ± 0.2         |
>     | **GaLore**             | 36G           | 82.8 ± 0.7           | 78.4 ± 0.2          | 55.8 ± 0.4          | 56.3 ± 0.5          | 79.0 ± 0.1          | 75.9 ± 0.4         | 46.2 ± 0.5         | 34.2 ± 0.1         | 63.6 ± 0.4         |
>     | **Our Method ($d=0.25\%$)** | 36G           | 85.0 ± 0.2           | 78.9 ± 0.2          | 55.7 ± 0.2          | 57.8 ± 0.1          | 78.9 ± 0.2          | 76.5 ± 0.2         | 47.3 ± 0.2         | 35.1 ± 0.3         | 64.4 ± 0.2         |
>     | **Our Method ($d=100\%$)** | 61G           | **87.1 ± 0.2**       | **79.5 ± 0.1**      | **58.3 ± 0.1**      | 58.1 ± 0.04         | **83.3 ± 0.2**      | **79.2 ± 0.2**     | **48.6 ± 0.1**     | **40.1 ± 0.2**     | **66.7 ± 0.1**     |
>
>
> **Comment 5: Baselines. How does this method compare with other memory-efficient approaches such as MeZO [1], SparseMeZO [2], and Extremely Sparse MeZO [3]?**
>
> - Thank you for your question. It is important to highlight that SPAM is a more general optimizer that works for both LLM pre-training and fine-tuning, whereas MeZO, SparseMeZO, and Extremely Sparse MeZO are primarily proposed for LLM fine-tuning.
>
> - To further address your concern, we conducted experiments comparing **SPAM** and **MeZO** in efficient LLM fine-tuning, using the OPT-1.3B pre-trained model. The results are summarized in the table below, with the training code obtained from [https://github.com/ZO-Bench/ZO-LLM](https://github.com/ZO-Bench/ZO-LLM). The results demonstrate that **SPAM** outperforms **MeZO** by a significant margin on both **WinoGrande** and **COPA** datasets.
>
>     **Table: Performance of MeZO and SPAM on WinoGrande and COPA Datasets** Experiments are based on the pre-trained OPT-1.3B model.
>
>     | **Methods**       | **WinoGrande** | **COPA** |
>     |--------------------|----------------|----------|
>     | **MeZO**          | 55.4           | 73.0     |
>     | **SPAM (d=0.25%)** | 58.3           | 75.0     |
>     | **SPAM (d=100%)**  | 59.4           | 79.0     |
>
> **Comment 6: Ablations:1) There is no comparison between Spike-Aware Clipping and simply nullifying gradient spikes.**
>
> - Thank you for your insightful comment. We conducted experiments on LLaMA-60M and LLaMA-130M to compare the performance of Spike-Aware Clipping and Nullifying Gradient Spikes. The results are presented both below and in the revised paper.
>
> - As shown in the table, SPAM with Spike-Aware Clipping outperforms SPAM with Nullifying, demonstrating the effectiveness of Spike-Aware Clipping.
>
>     **Table: Perplexity Comparison between SPAM with Spike-Aware Clipping and SPAM with Nullifying Gradient Spikes**
>
>     | Methods                       | LLaMA-60M | LLaMA-130M |
>     |-------------------------------|-----------|------------|
>     | SPAM w/ Spike-Aware Clipping | 30.46     | 23.36      |
>     | SPAM w/ Nullifying            | 30.86     | 23.62      |

---

> ### Author Response · Authors · 2024-11-21
> **Response to Reviewer BUJ8 (3/3)**
>
> **Comment 7: Ablations:2)It would be interesting to see if the parameter for sparse momentum could be selected in a structured manner, as described in [4].**
>
> - Thank you for your insightful suggestion. In response, we conducted experiments comparing SPAM with channel-wise sparse mask generation to SPAM with unstructured sparse mask generation. These experiments were carried out using LLaMA-60M and LLaMA-130M, with the results summarized in the table below.
>
> - As observed, SPAM with unstructured sparse masks significantly outperforms SPAM with channel-wise sparse masks. It is also important to note that, as long as the mask density remains the same, both SPAM with structured sparse masks and unstructured sparse masks consume the same amount of GPU memory since they store the same number of moments in the cache.
>
>     **Table: Perplexity of SPAM with Structural Sparse Momentum**
>
>     | Methods                          | LLaMA-60M          | LLaMA-130M         |
>     |----------------------------------|--------------------|--------------------|
>     | **Adam**                         | 34.06 (0.36G)      | 25.08 (0.76G)      |
>     | **SPAM with Unstructured Sparse Mask** | 30.46 (0.24G)      | 23.36 (0.52G)      |
>     | **SPAM with Channel Sparse Mask**     | 33.28 (0.24G)      | 25.92 (0.52G)      |
>
> **Comment 8: Ablations:3) What happens in the case $\Delta T < N$.**
>
> - Thank you for your question. We have included the performance results for cases where $\Delta T < N$ in the table below.
>
> - As shown in the table, reducing $\Delta T$ to less than the warm-up step $N$ negatively impacts performance. This is because, when $\Delta T < N$, the learning rate does not have sufficient time to reach its intended value (i.e., the original learning rate before the momentum reset), which adversely affects the performance of SPAM.
>
>     **Table: Perplexity with Various $\Delta T$ (Warmup Steps N=150, LLaMA-130M)**
>
>     | Intervals $\Delta T$           | Perplexity |
>     |--------------------------------|------------|
>     | 50 ($<$N)                      | 27.06      |
>     | 100 ($<$N)                     | 24.28      |
>     | 500 ($>$N) (ours)              | 23.36      |
>
> **Comment 9: Loss and Gradient plots of SPAM. How do the loss and gradient plots of SPAM compare with those of other methods shown in Figures 2-4??**
>
> - Thank you for your question. Following your suggestion,  we plotted the training loss and the number of gradient spikes for SPAM and Adam, with experiments conducted using the LLaMA-60M and LLaMA-1B models. Gradient spikes were identified using the condition \( GSS($g_i$) > 50 \). The results are now presented in Figure 10 of Appendix L in our revision.
>
> - The figure illustrates that SPAM effectively mitigates both training loss spikes and the occurrence of gradient spikes. This leads to a more stable and efficient training process.
>
>
> **Comment 10: Computational analysis. Can the authors report the computational overhead of SPAM compared to other methods?**
>
> - Thank you for your suggestion. To address this concern, we measured the running time per iteration for both LLaMA-60M and LLaMA-130M. The results, presented in the table below, indicate that SPAM incurs a slightly higher computational overhead compared to Adam, Adam-mini, and Adafactor. This overhead is primarily due to the gradient spike detection operation and the gradient selection based on sparse masks. However, we believe that such a small overhead is negligible compared to the overall pre-training time which can be dozens or hundreds of hours.
>
>     **Table: Running Time per Iteration (seconds).** The runtime is measured as the average of 100 iterations using one H100 GPU.
>
>     | Method           | Time per Iteration (LLaMA-60M) | Time per Iteration (LLaMA-130M) |
>     |-------------------|--------------------------------|---------------------------------|
>     | Adam             | 0.3666 (s)                    | 0.6397 (s)                     |
>     | Adam-mini        | 0.3614 (s)                    | 0.6472 (s)                     |
>     | Adafactor        | 0.3778 (s)                    | 0.6565 (s)                     |
>     | GaLore (rank=128)| 0.3871 (s)                    | 0.6702 (s)                     |
>     | SPAM (d=100%)    | 0.3814 (s)                    | 0.6683 (s)                     |
>     | SPAM (d=25%)     | 0.3799 (s)                    | 0.6658 (s)                     |

---

> > ### Comment · Reviewer_BUJ8 · 2024-11-24
> > **I have raised my score but there are some remaining concerns**
> >
> > I thank the authors for reading and answering my questions. I have raised my score to 6 but there are some remaining concerns.
> >
> > - Can the authors also compare Spike-Aware Clipping and simply nullifying gradient spikes during fine-tuning?
> > - Loss and Gradient plots of SPAM.
> >   - It looks like the training loss of Fig 10 left is different from Fig 2 right if I read it correctly.
> >   - In Line 386, the authors used $\theta = 5000$ but the spike detection was measured by $\theta = 50$.
> >   - How about the plots for fine-tuning, which is usually much “flatter” than pre-training?

---

> > > ### Author Response · Authors · 2024-11-27
> > > **Response to Reviewer BUJ8**
> > >
> > > Thank you for your feedback and for raising the rating.  We provide point-wise responses to address your concerns below.
> > >
> > > **Q1 Can the authors also compare Spike-Aware Clipping and simply nullifying gradient spikes during fine-tuning?**
> > >
> > > - Thank you for your question. We conducted experiments to fine-tune the pre-trained OPT-1.3B model on WinoGrande and COPA datasets respectively following the experimental setup in [1]. The experiments aimed to compare the performance of Spike-Aware Clipping and Nullifying Gradient Spikes in fine-tuning task. The results are presented both below and in the revised paper.
> > >
> > > - As shown in the table below, SPAM with Spike-Aware Clipping outperforms SPAM with Nullifying, further demonstrating the effectiveness of Spike-Aware Clipping.
> > >
> > >     **Table**: Comparison between SPAM w/ spike-aware clipping and SPAM w/ nullifying gradient spikes on fine-tuning task. The experiments are based on a pre-trained OPT-1.3B model.
> > >
> > >     | Methods                                    | WinoGrande | COPA |
> > >     |--------------------------------------------|------------|------|
> > >     | SPAM w/ Spike Aware Clipping (d=100%)      | 59.4       | 79.0 |
> > >     | SPAM w/ Spike Aware Clipping (d=0.25%)     | 58.3       | 75.0 |
> > >     | SPAM w/ Nullifying (d=100%)                | 58.0       | 78.0 |
> > >     | SPAM w/ Nullifying (d=0.25%)               | 57.4       | 75.0 |
> > >
> > > [1] Zhang, Yihua, et al. "Revisiting zeroth-order optimization for memory-efficient llm fine-tuning: A benchmark." ICML (2024).
> > >
> > > **Q2 It looks like the training loss of Fig 10 left is different from Fig 2 right if I read it correctly.?**
> > >
> > > - Thank you for your insightful comment and for bringing this to our attention. We sincerely appreciate your attention to the detail.
> > >
> > > - Upon double-checking the code, we identified that the discrepancy was caused by an issue in the loss collection process during multi-GPU training. Specifically, the experiments for Figure 2 were conducted on a single GPU, while those for Figure 10 were run using 4 GPUs for fast results, which introduced the inconsistency.
> > >
> > > - We have since fixed the bug and re-ran the experiments for Figure 10 (In **Appendix L**). The updated results now align with the numbers reported in Figure 2, resolving the issue. The updated figure demonstrates the same conclusion that SPAM effectively mitigates the loss spikes that occurred during training.
> > >
> > > **Q3 In Line 386, the authors used $\theta=5000$  but the spike detection  was measured by.$\theta=50$?**
> > >
> > > - Thanks for your question! When analyzing  Gradient Spikes in our paper, we define Gradient Spike Score (GSS) as $\mathrm{GSS}(g_i) = \frac{\lvert g_i \rvert}{\frac{1}{T+1} \sum_{j=0}^{T} \lvert g_j \rvert}$, and we set  $\theta=50$ for spike detection in this context. However, during training, we approximate the GSS using $\frac{g_i^2}{V_i}$ where $V_i=V_{i-1}*\beta_1+(1-\beta_1)*g_i^2$ is the moving average of $g_i^2$, to avoid the memory overhead required to store previous gradients. In this training setup, a threshold of $\theta=2500$ is approximately equivalent to $\theta=50$ as used in GSS.  **We have added a clarification in our latest revision to address this potential confusion**. Thank you for bringing this to our attention.
> > >
> > > **Q4 How about the plots for fine-tuning, which is usually much “flatter” than pre-training??**
> > >
> > > - We appreciate your interest in the fine-tuning scenario. And we have included the corresponding training curve in Figure 11 of **Appendix H** in our revision. It is worth noting that fine-tuning presents distinct dynamics compared to pre-training. In pre-training, the loss typically decreases consistently over the entire training process, reflecting gradual model improvement. In contrast, during fine-tuning, the loss often drops sharply in the initial steps, likely settling into a local basin, and then oscillates around it without further significant improvement.
> > >
> > > - Despite this behavior, SPAM achieves consistently lower loss than Adam, highlighting its efficacy even in this context. Our intuition is that SPAM may provide a more reasonable update direction by clipping noisy gradients for fine-tuning, preventing the optimizer from being guided to suboptimal solutions by abnormal gradients. While a more comprehensive investigation into these dynamics is certainly interesting, it is beyond the scope of this work, which primarily focuses on mitigating the negative impact of gradient spikes during pre-training. We acknowledge this as an exciting direction for future research and appreciate your suggestion!
> > >
> > > We appreciate your valuable feedback and hope we have addressed your concerns. Please let us know if you have any additional questions.
> > >
> > > Sincerely,
> > >
> > > Authors

---

> ### Comment · Reviewer_BUJ8 · 2024-11-29
> **Response to author**
>
> Thank you to the authors for addressing all my concerns. Since they have been resolved, I would like to maintain my acceptance score.

---

> > ### Author Response · Authors · 2024-11-30
> > **Thank you**
> >
> > We are delighted to hear that all your concerns have been addressed. Your valuable feedback has greatly contributed to the improvement of our work.
> > We appreciate your time and consideration in reviewing our submission and maintaining your acceptance score.
> >
> > Warm regards,
> >
> > The Authors

---

### Public Comment · ~Hengzhi_Pei1 · 2025-02-27
**Questions regarding nullifying the spiked gradients**

Hi, thanks for your nice work! I am interested in your experiments about nullifying the spiked gradients using GSS (Sec 2.2) and have some questions:
1. Is GSS calculated per element, i.e. maintain the history sum for each single element in the model weights?
2. For your experiment of nullifying the spiked gradients, do you use gradient clipping? If so, do you nullify before the clipping or after the clipping?
3. Do you further validate this finding in a larger LLM, 1B or 7B apart from LLama-60M?

I look forward to your reply!

---

> ### Public Comment · ~Tianjin_Huang1 · 2025-02-28
>
> Hi Pengzhi,
>
> Thank you for your questions.
>
> (1) Yes, GSS is calculated per element. We maintain the history sum when analyzing the gradient spikes. However, when using it for model training, we use the efficient version g**2/V as the metric.
>
> (2) We perform the nullifying operation without using gradient clipping.
>
> (3) We have not conducted the experiments on the 1B and 7B models yet.
>
> Best regards,

---

### Meta-Review · Area_Chair_ugyM · 2024-12-16

**Metareview:**

The paper tackles a practical  optimization method for improving the stability of large language model (LLM) training by mitigating gradient and loss spikes. SPAM achieves this through spike-aware gradient clipping and periodic momentum resets, coupled with a sparse momentum scheme to reduce memory usage. Empirical results on various LLMs (e.g., LLaMA, OPT, Pythia) and across pre-training and fine-tuning tasks demonstrate substantial improvements over Adam and other memory-efficient baselines, consistently reducing perplexity and enhancing stability. While some individual components (e.g. gradient clipping) per se are not new, the paper’s integration of these elements to address the spike problem is novel. Overall the paper demonstrates significant practical benefits, soundness, and general applicability for stabilizing LLM training.

**Additional Comments On Reviewer Discussion:**

During the rebuttal and discussion period, the reviewers requested additional baselines, more ablations, as well as clarifications on the relationship between spike-awareness and momentum resets, and the impact on memory. The authors addressed these concerns

---

### Decision · Program_Chairs · 2025-01-22

Accept (Poster)